# CoLiDE: Concomitant Linear DAG Estimation

**Seyed Saman Saboksayr, Gonzalo Mateos**
University of Rochester
{ssaboksa,gmateosb}@ur.rochester.edu

**Mariano Tepper**
Intel Labs
mariano.tepper@intel.com

## Abstract

We deal with the combinatorial problem of learning directed acyclic graph (DAG) structure from observational data adhering to a linear structural equation model (SEM). Leveraging advances in differentiable, nonconvex characterizations of acyclicity, recent efforts have advocated a continuous constrained optimization paradigm to efficiently explore the space of DAGs. Most existing methods employ lasso-type score functions to guide this search, which (i) require expensive penalty parameter retuning when the *unknown* SEM noise variances change across problem instances; and (ii) implicitly rely on limiting homoscedasticity assumptions. In this work, we propose a new convex score function for sparsity-aware learning of linear DAGs, which incorporates concomitant estimation of scale and thus effectively decouples the sparsity parameter from the exogenous noise levels. Regularization via a smooth, nonconvex acyclicity penalty term yields CoLiDE (**Co**ncomitant **Li**near **D**AG **E**stimation), a regression-based criterion amenable to efficient gradient computation and closed-form estimation of noise variances in heteroscedastic scenarios. Our algorithm outperforms state-of-the-art methods without incurring added complexity, especially when the DAGs are larger and the noise level profile is heterogeneous. We also find CoLiDE exhibits enhanced stability manifested via reduced standard deviations in several domain-specific metrics, underscoring the robustness of our novel linear DAG estimator.

## 1 Introduction

Directed acyclic graphs (DAGs) have well-appreciated merits for encoding causal relationships within complex systems, as they employ directed edges to link causes and their immediate effects. This graphical modeling framework has gained prominence in various machine learning (ML) applications spanning domains such as biology (Sachs et al., 2005; Lucas et al., 2004), genetics (Zhang et al., 2013), finance (Sanford & Moosa, 2012), and economics Pourret et al. (2008), to name a few. However, since the causal structure underlying a group of variables is often unknown, there is a need to address the task of inferring DAGs from observational data. While additional interventional data are provably beneficial to the related problem of *causal discovery* (Lippe et al., 2021; Squires et al., 2020; Addanki et al., 2020; Brouillard et al., 2020), said interventions may be infeasible or ethically challenging to implement. Learning a DAG solely from observational data poses significant computational challenges, primarily due to the combinatorial acyclicity constraint which is notoriously difficult to enforce (Chickering, 1996; Chickering et al., 2004). Moreover, distinguishing between DAGs that generate the same observational data distribution is nontrivial. This identifiability challenge may arise when data are limited (especially in high-dimensional settings), or, when candidate graphs in the search space exhibit so-termed Markov equivalence; see e.g., (Peters et al., 2017).

Recognizing that DAG learning from observational data is in general an NP-hard problem, recent efforts have advocated a continuous relaxation approach which offers an efficient means of exploring the space of DAGs (Zheng et al., 2018; Ng et al., 2020; Bello et al., 2022). In addition to breakthroughs in differentiable, nonconvex characterizations of acyclicity (see Section 3), the choice of an appropriate score function is paramount to guide continuous optimization techniques that search for faithful representations of the underlying DAG. Likelihood-based methods (Ng et al., 2020) enjoy desirable statistical properties, provided that the postulated probabilistic model aligns with the actual causal relationships (Hastie et al., 2009; Casella & Berger, 2021). On the other hand, regression-based methods (Zheng et al., 2018; Bello et al., 2022) exhibit computational efficiency,

robustness, and even consistency (Loh & Bühlmann, 2014), especially in high-dimensional settings where both data scarcity and model uncertainty are prevalent (Ndiaye et al., 2017). Still, workhorse regression-based methods such as ordinary least squares (LS) rely on the assumption of homoscedasticity, meaning the variances of the exogenous noises are identical across variables. Deviations from this assumption can introduce biases in standard error estimates, hindering the accuracy of causal discovery (Long & Ervin, 2000; Loh & Bühlmann, 2014). Fundamentally, for heteroscedastic linear Gaussian models the DAG structure is non-identifiable from observational data (Peters et al., 2017).

Score functions often include a sparsity-promoting regularization, for instance an $\ell_1$-norm penalty as in lasso regression (Tibshirani, 1996). This in turn necessitates careful fine-tuning of the penalty parameter that governs the trade-off between sparsity and data fidelity (Massias et al., 2018). Theoretical insights suggest an appropriately scaled regularization parameter proportional to the observation noise level (Bickel et al., 2009), but the latter is typically unknown. Accordingly, several linear regression studies (unrelated to DAG estimation) have proposed convex *concomitant* estimators based on scaled LS, which jointly estimate the noise level along with the regression coefficients (Owen, 2007; Sun & Zhang, 2012; Belloni et al., 2011; Ndiaye et al., 2017). A noteworthy representative is the smoothed concomitant lasso (Ndiaye et al., 2017), which not only addresses the aforementioned parameter fine-tuning predicament but also accommodates heteroscedastic linear regression scenarios where noises exhibit non-equal variances. While these challenges are not extraneous to DAG learning, the impact of concomitant estimators is so far largely unexplored in this timely field.

**Contributions.** In this work, we bring to bear ideas from concomitant scale estimation in (sparse) linear regression and propose a novel convex score function for regression-based inference of linear DAGs, demonstrating significant improvements relative to existing state-of-the-art methods.[1] Our contributions can be summarized as follows:

• We propose a new convex score function for sparsity-aware learning of linear DAGs, which incorporates concomitant estimation of scale parameters to enhance DAG topology inference using continuous first-order optimization. We augment this LS-based score function with a smooth, nonconvex acyclicity penalty term to arrive at CoLiDE (**Co**ncomitant **Li**near **D**AG **E**stimation), a simple regression-based criterion that facilitates *efficient* computation of gradients and estimation of exogenous noise levels via closed-form expressions. To the best of our knowledge, this is the first time that ideas from concomitant scale estimation permeate benefits to DAG learning.

• In existing methods, the sparsity regularization parameter depends on the unknown exogenous noise levels, making the calibration challenging. CoLiDE effectively removes this coupling, leading to minimum (or no) recalibration effort across diverse problem instances. Our score function is fairly robust to deviations from Gaussianity, and relative to ordinary LS used in prior score-based DAG estimators, experiments show CoLiDE is more suitable for challenging heteroscedastic scenarios.

• We demonstrate CoLiDE's effectiveness through comprehensive experiments with both simulated (linear SEM) and real-world datasets. When benchmarked against state-of-the-art DAG learning algorithms, CoLiDE consistently attains better recovery performance across graph ensembles and exogenous noise distributions, especially when the DAGs are larger and the noise level profile is heterogeneous. We also find CoLiDE exhibits enhanced stability manifested via reduced standard deviations in various domain-specific metrics, underscoring the robustness of our novel estimator.

## 2 PRELIMINARIES AND PROBLEM STATEMENT

Consider a directed graph $\mathcal{G}(\mathcal{V}, \mathcal{E}, \mathbf{W})$, where $\mathcal{V} = \{1, \dots, d\}$ represents the set of vertices, and $\mathcal{E} \subseteq \mathcal{V} \times \mathcal{V}$ is the set of edges. The adjacency matrix $\mathbf{W} = [\mathbf{w}_1, \dots, \mathbf{w}_d] \in \mathbb{R}^{d \times d}$ collects the edge weights, with $W_{ij} \neq 0$ indicating a direct link from node $i$ to node $j$. We henceforth assume that $\mathcal{G}$ belongs to the space $\mathbb{D}$ of DAGs, and rely on the graph to represent conditional independencies among the variables in the random vector $\mathbf{x} = [x_1, \dots, x_d]^\top \in \mathbb{R}^d$. Indeed, if the joint distribution $\mathbb{P}(\mathbf{x})$ satisfies a Markov property with respect to $\mathcal{G} \in \mathbb{D}$, each random variable $x_i$ depends solely on

---

[1]While preparing the final version of this manuscript, we became aware of independent work in the unpublished preprint (Xue et al., 2023), which also advocates estimating (and accounting for) exogenous noise variances to improve DAG topology inference. However, `dotears` is a two-step marginal estimator requiring *additional interventional* data, while our concomitant framework leads to a new LS-based score function and, importantly, relies on observational data only.

its parents $\mathrm{PA}_i = \{j \in \mathcal{V} : W_{ji} \neq 0\}$. Here, we focus on *linear* structural equation models (SEMs) to generate said probability distribution, whereby the relationship between each random variable and its parents is given by $x_i = \mathbf{w}_i^\top \mathbf{x} + z_i, \forall i \in \mathcal{V}$, and $\mathbf{z} = [z_1, \ldots, z_d]^\top$ is a vector of mutually independent, exogenous noises; see e.g., (Peters et al., 2017). Noise variables $z_i$ can have different variances and need not be Gaussian distributed. For a dataset $\mathbf{X} \in \mathbb{R}^{d \times n}$ of $n$ i.i.d. samples drawn from $\mathbb{P}(\mathbf{x})$, the linear SEM equations can be written in compact matrix form as $\mathbf{X} = \mathbf{W}^\top \mathbf{X} + \mathbf{Z}$.

**Problem statement.** Given the data matrix $\mathbf{X}$ adhering to a linear SEM, our goal is to learn the latent DAG $\mathcal{G} \in \mathbb{D}$ by estimating its adjacency matrix $\mathbf{W}$ as the solution to the optimization problem

$$\min_{\mathbf{W}} \quad \mathcal{S}(\mathbf{W}) \quad \text{subject to} \quad \mathcal{G}(\mathbf{W}) \in \mathbb{D}, \tag{1}$$

where $\mathcal{S}(\mathbf{W})$ is a data-dependent score function to measure the quality of the candidate DAG. Irrespective of the criterion, the non-convexity comes from the combinatorial acyclicity constraint $\mathcal{G}(\mathbf{W}) \in \mathbb{D}$; see also Appendix A for a survey of combinatorial search approaches relevant to solving (1), but less related to the continuous relaxation methodology pursued here (see Section 3).

A proper score function typically encompasses a loss or data fidelity term ensuring alignment with the SEM as well as regularizers to promote desired structural properties on the sought DAG. For a linear SEM, the ordinary LS loss $\frac{1}{2n}\|\mathbf{X} - \mathbf{W}^\top \mathbf{X}\|_F^2$ is widely adopted, where $\|\cdot\|_F$ is the Frobenius norm. When the exogenous noises are Gaussian, the data log-likelihood can be an effective alternative (Ng et al., 2020). Since sparsity is a cardinal property of most real-world graphs, it is prudent to augment the loss with an $\ell_1$-norm regularizer to yield $\mathcal{S}(\mathbf{W}) = \frac{1}{2n}\|\mathbf{X} - \mathbf{W}^\top \mathbf{X}\|_F^2 + \lambda\|\mathbf{W}\|_1$, where $\lambda \geq 0$ is a tuning parameter that controls edge sparsity. The score function $\mathcal{S}(\mathbf{W})$ bears resemblance to the *multi-task* variant of lasso regression (Tibshirani, 1996), specifically when the response and design matrices coincide. Optimal rates for lasso hinge on selecting $\lambda \asymp \sigma\sqrt{\log d/n}$ (Bickel et al., 2009; Li et al., 2020). However, the exogenous noise variance $\sigma^2$ is rarely known in practice. This challenge is compounded in heteroscedastic settings, where one should adopt a *weighted* LS score (see (Loh & Bühlmann, 2014) for an exception unlike most DAG learning methods that stick to bias-inducing ordinary LS). Recognizing these limitations, in Section 4 we propose a novel LS-based score function to facilitate joint estimation of the DAG and the noise levels.

## 3 RELATED WORK

We briefly review differentiable optimization approaches that differ in how they handle the acyclicity constraint, namely by using an explicit DAG parameterization or continuous relaxation techniques.

**Continuous relaxation.** Noteworthy methods advocate an exact acyclicity characterization using nonconvex, smooth functions $\mathcal{H} : \mathbb{R}^{d \times d} \mapsto \mathbb{R}$ of the adjacency matrix, whose zero level set is $\mathbb{D}$. One can thus relax the combinatorial constraint $\mathcal{G}(\mathbf{W}) \in \mathbb{D}$ by instead enforcing $\mathcal{H}(\mathbf{W}) = 0$, and tackle the DAG learning problem using standard continuous optimization algorithms (Zheng et al., 2018; Yu et al., 2019; Wei et al., 2020; Bello et al., 2022). The pioneering NOTEARS formulation introduced $\mathcal{H}_{\text{expm}}(\mathbf{W}) = \mathrm{Tr}(e^{\mathbf{W} \circ \mathbf{W}}) - d$, where $\circ$ denotes the Hadamard (element-wise) product and $\mathrm{Tr}(\cdot)$ is the matrix trace operator (Zheng et al., 2018). Diagonal entries of powers of $\mathbf{W} \circ \mathbf{W}$ encode information about cycles in $\mathcal{G}$, hence the suitability of the chosen function. Follow-up work suggested a more computationally efficient acyclicity function $\mathcal{H}_{\text{poly}}(\mathbf{W}) = \mathrm{Tr}((\mathbf{I} + \frac{1}{d}\mathbf{W} \circ \mathbf{W})^d) - d$, where $\mathbf{I}$ is the identity matrix (Yu et al., 2019); see also (Wei et al., 2020) that studies the general family $\mathcal{H}(\mathbf{W}) = \sum_{k=1}^d c_k \mathrm{Tr}((\mathbf{W} \circ \mathbf{W})^d)$, $c_k > 0$. Recently, Bello et al. (2022) proposed the log-determinant acyclicity characterization $\mathcal{H}_{\text{ldet}}(\mathbf{W}, s) = d\log(s) - \log(\det(s\mathbf{I} - \mathbf{W} \circ \mathbf{W}))$, $s \in \mathbb{R}$; outperforming prior relaxation methods in terms of (nonlinear) DAG recovery and efficiency. Although global optimality results are so far elusive, progress is being made (Deng et al., 2023b).

**Order-based methods.** Other recent approaches exploit the neat equivalence $\mathcal{G}(\mathbf{W}) \in \mathbb{D} \Leftrightarrow \mathbf{W} = \mathbf{\Pi}^\top \mathbf{U} \mathbf{\Pi}$, where $\mathbf{\Pi} \in \{0, 1\}^{d \times d}$ is a permutation matrix (essentially encoding the variables' causal ordering) and $\mathbf{U} \in \mathbb{R}^{d \times d}$ is an upper-triangular weight matrix. Consequently, one can search over exact DAGs by formulating an end-to-end differentiable optimization framework to minimize $\mathcal{S}(\mathbf{\Pi}^\top \mathbf{U} \mathbf{\Pi})$ jointly over $\mathbf{\Pi}$ and $\mathbf{U}$, or in two steps. Even for nonlinear SEMs, the challenging process of determining the appropriate node ordering is typically accomplished through the Birkhoff polytope of permutation matrices, utilizing techniques like the Gumbel-Sinkhorn approximation (Cundy et al., 2021), or, the SoftSort operator (Charpentier et al., 2022). Limitations stemming from

misaligned forward and backward passes that respectively rely on hard and soft permutations are well documented (Zantedeschi et al., 2023). The DAGuerreotype approach in (Zantedeschi et al., 2023) instead searches over the Permutahedron of *vector* orderings and allows for non-smooth edge weight estimators. TOPO (Deng et al., 2023a) performs a bi-level optimization, relying on topological order swaps at the outer level. Still, optimization challenges towards accurately recovering the causal ordering remain, especially when data are limited and the noise level profile is heterogeneous.

While this work is framed within the continuous constrained relaxation paradigm to linear DAG learning, preliminary experiments with TOPO (Deng et al., 2023a) show that CoLiDE also benefits order-based methods (see Appendix E.8). We leave a full exploration as future work.

## 4    CONCOMITANT LINEAR DAG ESTIMATION

Going back to our discussion on lasso-type score functions in Section 2, minimizing $\mathcal{S}(\mathbf{W}) = \frac{1}{2n}\|\mathbf{X} - \mathbf{W}^\top\mathbf{X}\|_F^2 + \lambda\|\mathbf{W}\|_1$ subject to a smooth acyclicity constraint $\mathcal{H}(\mathbf{W}) = 0$ (as in e.g., NoTears (Zheng et al., 2018)): (i) requires carefully retuning $\lambda$ when the unknown SEM noise variance changes across problem instances; and (ii) implicitly relies on limiting homoscedasticity assumptions due to the ordinary LS loss. To address issues (i)-(ii), here we propose a new convex score function for linear DAG estimation that incorporates concomitant estimation of scale. This way, we obtain a procedure that is robust (both in terms of DAG estimation performance and parameter fine-tuning) to possibly heteroscedastic exogenous noise profiles. We were inspired by the literature of concomitant scale estimation in sparse linear regression (Owen, 2007; Belloni et al., 2011; Sun & Zhang, 2012; Ndiaye et al., 2017); see Appendix A.1 for concomitant lasso background that informs the approach pursued here. Our method is dubbed CoLiDE (**Co**ncomitant **Li**near **D**AG **E**stimation).

**Homoscedastic setting.** We start our exposition with a simple scenario whereby all exogenous variables $z_1, \ldots, z_d$ in the linear SEM have identical variance $\sigma^2$. Building on the smoothed concomitant lasso (Ndiaye et al., 2017), we formulate the problem of jointly estimating the DAG adjacency matrix $\mathbf{W}$ and the exogenous noise scale $\sigma$ as

$$\min_{\mathbf{W}, \sigma \geq \sigma_0} \underbrace{\frac{1}{2n\sigma}\|\mathbf{X} - \mathbf{W}^\top\mathbf{X}\|_F^2 + \frac{d\sigma}{2} + \lambda\|\mathbf{W}\|_1}_{:=\mathcal{S}(\mathbf{W},\sigma)} \quad \text{subject to} \quad \mathcal{H}(\mathbf{W}) = 0, \tag{2}$$

where $\mathcal{H} : \mathbb{R}^{d \times d} \mapsto \mathbb{R}$ is a nonconvex, smooth function, whose zero level set is $\mathbb{D}$ as discussed in Section 3. Notably, the weighted, regularized LS score function $\mathcal{S}(\mathbf{W}, \sigma)$ is now also a function of $\sigma$, and it can be traced back to the robust linear regression work of Huber (1981). Due to the rescaled residuals, $\lambda$ in (2) decouples from $\sigma$ as minimax optimality now requires $\lambda \asymp \sqrt{\log d/n}$ (Li et al., 2020; Belloni et al., 2011). A minor tweak to the argument in the proof of (Owen, 2007, Theorem 1) suffices to establish that $\mathcal{S}(\mathbf{W}, \sigma)$ is jointly convex in $\mathbf{W}$ and $\sigma$. Of course, (2) is still a nonconvex optimization problem by virtue of the acyclicity constraint $\mathcal{H}(\mathbf{W}) = 0$. Huber (1981) included the term $d\sigma/2$ so that the resulting squared scale estimator is consistent under Gaussianity. The additional constraint $\sigma \geq \sigma_0$ safeguards against potential ill-posed scenarios where the estimate $\hat{\sigma}$ approaches zero. Following the guidelines in (Ndiaye et al., 2017), we set $\sigma_0 = \frac{\|\mathbf{X}\|_F}{\sqrt{dn}} \times 10^{-2}$.

With regards to the choice of the acyclicity function, we select $\mathcal{H}_{\text{ldet}}(\mathbf{W}, s) = d\log(s) - \log(\det(s\mathbf{I} - \mathbf{W} \circ \mathbf{W}))$ based on its more favorable gradient properties in addition to several other compelling reasons outlined in (Bello et al., 2022, Section 3.2). Moreover, while LS-based linear DAG learning approaches are prone to introducing cycles in the estimated graph, it was noted that a log-determinant term arising with the Gaussian log-likelihood objective tends to mitigate this undesirable effect (Ng et al., 2020). Interestingly, the same holds true when $\mathcal{H}_{\text{ldet}}$ is chosen as a regularizer, but this time without being tied to Gaussian assumptions. Before moving on to optimization issues, we emphasize that our approach is in principle flexible to accommodate other convex loss functions beyond LS (e.g., Huber's loss for robustness against heavy-tailed contamination), other acyclicity functions, and even nonlinear SEMs parameterized using e.g., neural networks. All these are interesting CoLiDE generalizations beyond the scope of this paper.

**Optimization considerations.** Motivated by our choice of the acyclicity function, to solve the constrained optimization problem (2) we follow the DAGMA methodology by Bello et al. (2022). Therein, it is suggested to solve a sequence of unconstrained problems where $\mathcal{H}_{\text{ldet}}$ is dualized and

viewed as a regularizer. This technique has proven more effective in practice for our specific problem as well, when compared to, say, an augmented Lagrangian method. Given a decreasing sequence of values $\mu_k \to 0$, at step $k$ of the COLIDE-EV (equal variance) algorithm one solves

$$\min_{\mathbf{W}, \sigma \geq \sigma_0} \mu_k \left[ \frac{1}{2n\sigma} \|\mathbf{X} - \mathbf{W}^\top \mathbf{X}\|_F^2 + \frac{d\sigma}{2} + \lambda \|\mathbf{W}\|_1 \right] + \mathcal{H}_{\mathrm{ldet}}(\mathbf{W}, s_k), \tag{3}$$

where the schedule of hyperparameters $\mu_k \geq 0$ and $s_k > 0$ must be prescribed prior to implementation; see Section 4.1. Decreasing the value of $\mu_k$ enhances the influence of the acyclicity function $\mathcal{H}_{\mathrm{ldet}}(\mathbf{W}, s)$ in the objective. Bello et al. (2022) point out that the sequential procedure (3) is reminiscent of the central path approach of barrier methods, and the limit of the central path $\mu_k \to 0$ is guaranteed to yield a DAG. In theory, this means no additional post-processing (e.g., edge trimming) is needed. However, in practice we find some thresholding is required to reduce false positives.

Unlike Bello et al. (2022), CoLiDE-EV jointly estimates the noise level $\sigma$ and the adjacency matrix $\mathbf{W}$ for each $\mu_k$. To this end, one could solve for $\sigma$ in closed form [cf. (4)] and plug back the solution in (3), to obtain a DAG-regularized square-root lasso (Belloni et al., 2011) type of problem in $\mathbf{W}$. We did not follow this path since the resulting loss $\|\mathbf{X} - \mathbf{W}^\top \mathbf{X}\|_F$ is not decomposable across samples, challenging mini-batch based stochastic optimization if it were needed for scalability (in Appendix B, we show that our score function is fully decomposable and present corresponding preliminary experiments). A similar issue arises with GOLEM (Ng et al., 2020), where the derived Gaussian profile likelihood yields a non-separable log-sum loss. Alternatively, and similar to the smooth concomitant lasso algorithm Ndiaye et al. (2017), we rely on (inexact) block coordinate descent (BCD) iterations. This cyclic strategy involves fixing $\sigma$ to its most up-to-date value and minimizing (3) inexactly w.r.t. $\mathbf{W}$, subsequently updating $\sigma$ in closed form given the latest $\mathbf{W}$ via

$$\hat{\sigma} = \max \left( \sqrt{\mathrm{Tr}\left( (\mathbf{I} - \mathbf{W})^\top \mathrm{cov}(\mathbf{X})(\mathbf{I} - \mathbf{W}) \right)/d}, \sigma_0 \right), \tag{4}$$

where $\mathrm{cov}(\mathbf{X}) := \frac{1}{n}\mathbf{X}\mathbf{X}^\top$ is the precomputed sample covariance matrix. The mutually-reinforcing interplay between noise level and DAG estimation should be apparent. There are several ways to inexactly solve the $\mathbf{W}$ subproblem using first-order methods. Given the structure of (3), an elegant solution is to rely on the proximal linear approximation. This leads to an ISTA-type update for $\mathbf{W}$, and overall a provably convergent BCD procedure in this nonsmooth, nonconvex setting (Yang et al., 2020, Theorem 1). Because the required line search can be computationally taxing, we opt for a simpler heuristic which is to run a single step of the ADAM optimizer (Kingma & Ba, 2015) to refine $\mathbf{W}$. We observed that running multiple ADAM iterations yields marginal gains, since we are anyways continuously re-updating $\mathbf{W}$ in the BCD loop. This process is repeated until either convergence is attained, or, a prescribed maximum iteration count $T_k$ is reached. Additional details on gradient computation of (3) w.r.t. $\mathbf{W}$ and the derivation of (4) can be found in Appendix B.

**Heteroscedastic setting.** We also address the more challenging endeavor of learning DAGs in heteroscedastic scenarios, where noise variables have non-equal variances (NV) $\sigma_1^2, \ldots, \sigma_d^2$. Bringing to bear ideas from the generalized concomitant multi-task lasso (Massias et al., 2018) and mimicking the optimization approach for the EV case discussed earlier, we propose the CoLiDE-NV estimator

$$\min_{\mathbf{W}, \boldsymbol{\Sigma} \geq \boldsymbol{\Sigma}_0} \mu_k \left[ \frac{1}{2n} \mathrm{Tr}\left( (\mathbf{X} - \mathbf{W}^\top \mathbf{X})^\top \boldsymbol{\Sigma}^{-1} (\mathbf{X} - \mathbf{W}^\top \mathbf{X}) \right) + \frac{1}{2}\mathrm{Tr}(\boldsymbol{\Sigma}) + \lambda \|\mathbf{W}\|_1 \right] + \mathcal{H}_{\mathrm{ldet}}(\mathbf{W}, s_k).$$
$$\tag{5}$$

Note that $\boldsymbol{\Sigma} = \mathrm{diag}(\sigma_1, \ldots, \sigma_d)$ is a diagonal matrix of exogenous noise *standard deviations* (hence not a covariance matrix). Once more, we set $\boldsymbol{\Sigma}_0 = \sqrt{\mathrm{diag}\left(\mathrm{cov}(\mathbf{X})\right)} \times 10^{-2}$, where $\sqrt{(\cdot)}$ is meant to be taken element-wise. A closed form solution for $\boldsymbol{\Sigma}$ given $\mathbf{W}$ is also readily obtained,

$$\hat{\boldsymbol{\Sigma}} = \max \left( \sqrt{\mathrm{diag}\left( (\mathbf{I} - \mathbf{W})^\top \mathrm{cov}(\mathbf{X})(\mathbf{I} - \mathbf{W}) \right)}, \boldsymbol{\Sigma}_0 \right). \tag{6}$$

A summary of the overall computational procedure for both CoLiDE variants is tabulated under Algorithm 1; see Appendix B for detailed gradient expressions and a computational complexity discussion. CoLiDE's per iteration cost is $\mathcal{O}(d^3)$, on par with state-of-the-art DAG learning methods.

The first summand in (5) resembles the consistent weighted LS estimator studied in Loh & Bühlmann (2014), but therein the assumption is that exogenous noise variances are known up to

---

**Algorithm 1:** CoLiDE optimization

**In:** data $\mathbf{X}$ and hyperparameters $\lambda$ and $H = \{(\mu_k, s_k, T_k)\}_{k=1}^K$.
**Out:** DAG $\mathbf{W}$ and the noise estimate $\sigma$ (EV) or $\mathbf{\Sigma}$ (NV).
Compute lower-bounds $\sigma_0$ or $\mathbf{\Sigma}_0$.
Initialize $\mathbf{W} = \mathbf{0}$, $\sigma = \sigma_0 \times 10^2$ or $\mathbf{\Sigma} = \mathbf{\Sigma}_0 \times 10^2$.
**foreach** $(\mu_k, s_k, T_k) \in H$ **do**
  **for** $t = 1, \ldots, T_k$ **do**
    Apply CoLiDE-EV or NV updates using $\mu_k$ and $s_k$.

**Function** *CoLiDE-EV update*:
  Update $\mathbf{W}$ with one iteration of
  a first-order method for (3)
  Compute $\hat{\sigma}$ using (4)
**Function** *CoLiDE-NV update*:
  Update $\mathbf{W}$ with one iteration of
  a first-order method for (5)
  Compute $\hat{\mathbf{\Sigma}}$ using (6)

---

a constant factor. CoLiDE-NV removes this assumption by jointly estimating $\mathbf{W}$ and $\mathbf{\Sigma}$, with marginal added complexity over finding the DAG structure alone. Like GOLEM (Ng et al., 2020) and for general (non-identifiable) linear Gaussian SEMs, as $n \to \infty$ CoLiDE-NV probably yields a DAG that is quasi-equivalent to the ground truth graph (see Appendix C for futher details).

### 4.1 FURTHER ALGORITHMIC DETAILS

**Initialization.** Following Bello et al. (2022), we initialize $\mathbf{W} = \mathbf{0}$ as it always lies in the feasibility region of $\mathcal{H}_{\text{ldet}}$. We also initialize $\sigma = \sigma_0 \times 10^2$ in CoLiDE-EV or $\mathbf{\Sigma} = \mathbf{\Sigma}_0 \times 10^2$ for CoLiDE-NV.

**Hyperparameter selection and termination rule.** To facilitate the comparison with the state-of-the-art DAGMA and to better isolate the impact of our novel score function, we use the hyperparameters selected by Bello et al. (2022) for CoLiDE-EV and NV. Hence, our algorithm uses $\lambda = 0.05$ and employs $K = 4$ decreasing values of $\mu_k \in [1.0, 0.1, 0.01, 0.001]$, and the maximum number of BCD iterations is specified as $T_k = [2 \times 10^4, 2 \times 10^4, 2 \times 10^4, 7 \times 10^4]$. Furthermore, early stopping is incorporated, activated when the relative error between consecutive values of the objective function falls below $10^{-6}$. The learning rate for ADAM is $3 \times 10^{-4}$. We adopt $s_k \in [1, 0.9, 0.8, 0.7]$.

**Post-processing.** Similar to several previous works (Zheng et al., 2018; Ng et al., 2020; Bello et al., 2022), we conduct a final thresholding step to reduce false discoveries. In this post-processing step, edges with absolute weights smaller than $0.3$ are removed.

## 5 EXPERIMENTAL RESULTS

We now show that CoLiDE leads to state-of-the-art DAG estimation results by conducting a comprehensive evaluation against other state-of-the-art approaches: GES (Ramsey et al., 2017), GOLEM (Ng et al., 2020), DAGMA (Bello et al., 2022), SortNRegress (Reisach et al., 2021), and DAGuerreotype (Zantedeschi et al., 2023). We omit conceptually important methods like NoTears (Zheng et al., 2018), that have been outclassed in practice. To improve the visibility of the figures, we report in each experiment the results of the best performing methods, excluding those that perform particularly poorly. We use standard DAG recovery metrics: (normalized) Structural Hamming Distance (SHD), SHD over the Markov equivalence class (SHD-C), Structural Intervention Distance (SID), True Positive Rate (TPR), and False Discovery Rate (FDR). We also assess CoLiDE's ability to estimate the noise level(s) across different scenarios. Further details about the setup are in Appendix D.

**DAG generation.** As standard (Zheng et al., 2018), we generate the ground truth DAGs utilizing the Erdős-Rényi or the Scale-Free random models, respectively denoted as ER$k$ or SF$k$, where $k$ is the average nodal degree. Edge weights for these DAGs are drawn uniformly from a range of feasible edge weights. We present results for $k = 4$, the most challenging setting (Bello et al., 2022).

**Data generation.** Employing the linear SEM, we simulate $n = 1000$ samples using the homo- or heteroscedastic noise models and drawing from diverse noise distributions, i.e., Gaussian, Exponential, and Laplace (see Appendix D). Unless indicated otherwise, we report the aggregated results of ten independent runs by repeating all experiments 10 times, each with a distinct DAG.

Appendix E contains additional results (e.g., other graph types, different number of nodes, and $k$ configurations). There, CoLiDE prevails with similar observations as in the cases discussed next.

**Homoscedastic setting.** We begin by assuming equal noise variances across all nodes. We employ ER4 and SF4 graphs with 200 nodes and edge weights drawn uniformly from the range $\mathcal{E} \in [-2, -0.5] \cup [0.5, 2]$ (Zheng et al., 2018; Ng et al., 2020; Bello et al., 2022).

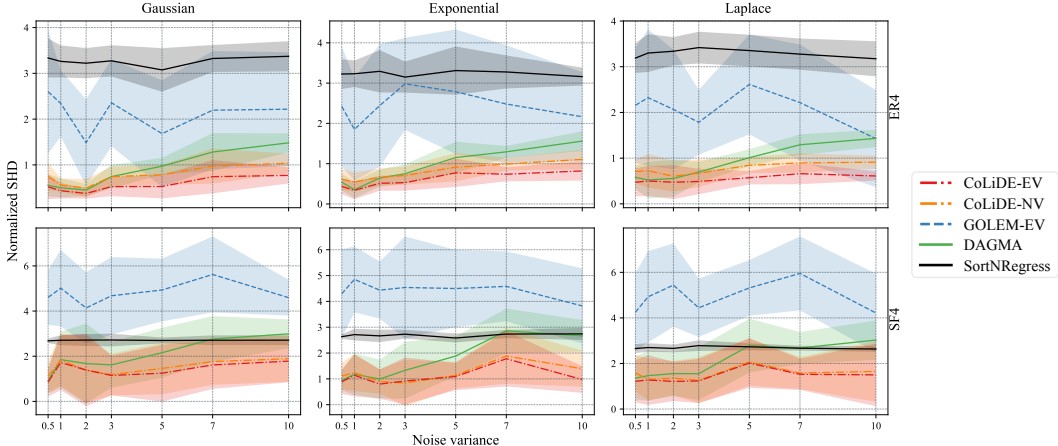

Figure 1: Mean DAG recovery performance, plus/minus one standard deviation, is evaluated for ER4 (top row) and SF4 (bottom row) graphs, each with 200 nodes, assuming equal noise variances. Each column corresponds to a different noise distribution.

Table 1: DAG recovery results for 200-node ER4 graphs under homoscedastic Gaussian noise.

|  | Noise variance = 1.0 | | | | Noise variance = 5.0 | | | |
|---|---|---|---|---|---|---|---|---|
|  | GOLEM | DAGMA | CoLiDE-NV | CoLiDE-EV | GOLEM | DAGMA | CoLiDE-NV | CoLiDE-EV |
| SHD | 468.6±144.0 | 100.1±41.8 | 111.9±29 | **87.3±33.7** | 336.6±233.0 | 194.4±36.2 | 157±44.2 | **105.6±51.5** |
| SID | 22260±3951 | 4389±1204 | 5333±872 | **4010±1169** | 14472±9203 | 6582±1227 | 6067±1088 | **4444±1586** |
| SHD-C | 473.6±144.8 | 101.2±41.0 | 113.6±29.2 | **88.1±33.8** | 341.0±234.9 | 199.9±36.1 | 161.0±43.5 | **107.1±51.6** |
| FDR | 0.28±0.10 | 0.07±0.03 | 0.08±0.02 | **0.06±0.02** | 0.21±0.13 | 0.15±0.02 | 0.12±0.03 | **0.08±0.04** |
| TPR | 0.66±0.09 | 0.94±0.01 | 0.93±0.01 | **0.95±0.01** | 0.76±0.18 | 0.92±0.01 | 0.93±0.01 | **0.95±0.01** |

In Figure 1, we investigate the impact of noise levels varying from 0.5 to 10 for different noise distributions. CoLiDE-EV clearly outperforms its competitors, consistently reaching a lower SHD. Here, it is important to highlight that GOLEM, being based on the profile log-likelihood for the Gaussian case, intrinsically addresses noise estimation for that particular scenario. However, the more general CoLiDE formulation still exhibits superior performance even in GOLEM's specific scenario (left column). We posit that the logarithmic nonlinearity in GOLEM's data fidelity term hinders its ability to fit the data. CoLiDE's noise estimation provides a more precise correction, equivalent to a square-root nonlinearity [see (9) in Appendix A.1], giving more weight to the data fidelity term and consequently allowing to fit the data more accurately.

In Table 1, we deep-dive in two particular scenarios: (1) when the noise variance equals 1, as in prior studies (Zheng et al., 2018; Ng et al., 2020; Bello et al., 2022), and (2) when the noise level is increased to 5. Note that SortNRegress does not produce state-of-the-art results (SHD of 652.5 and 615 in each case, respectively), which relates to the non-triviality of the problem instance. These cases show that CoLiDE's advantage over its competitors is not restricted to SHD alone, but equally extends to all other relevant metrics. This behavior is accentuated when the noise variance is set to 5, as CoLiDE naturally adapts to different noise regimes without any manual tuning of its hyperparameters. Additionally, CoLiDE-EV consistently yields lower standard deviations than the alternatives across all metrics, underscoring its robustness.

CoLiDE-NV, although overparametrized for homoscedastic problems, performs remarkably well, either being on par with CoLiDE-EV or the second-best alternative in Figure 1 and Table 1. This is of particular importance as the characteristics of the noise are usually unknown in practice, favoring more general and versatile formulations.

**Heteroscedastic setting.** The heteroscedastic scenario, where nodes do not share the same noise variance, presents further challenges. Notably, this setting is known to be non-identifiable (Ng et al., 2020) from observational data. This issue is exacerbated as the number $d$ of nodes grows as, whether we are estimating the variances explicitly or implicitly, the problem contains $d$ additional unknowns, which renders its optimization harder from a practical perspective. We select the edge weights by uniformly drawing from $[-1, -0.25] \cup [0.25, 1]$ and the noise variance of each node from $[0.5, 10]$.

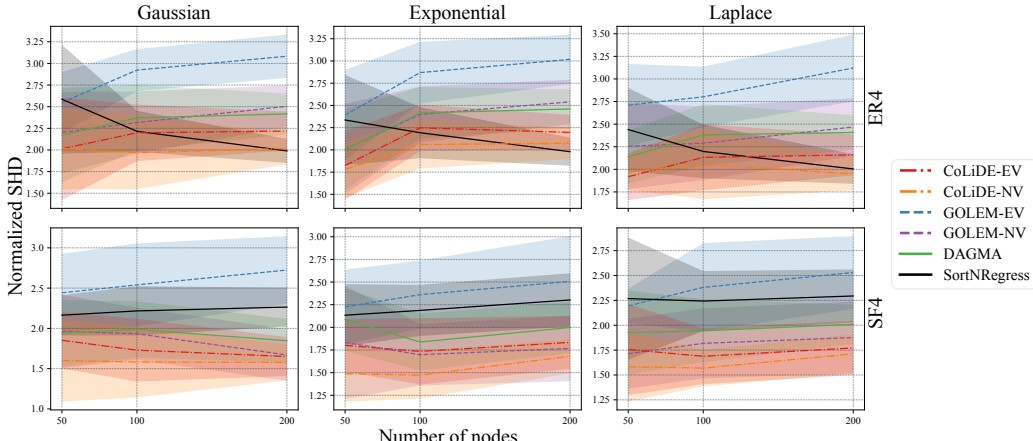

Figure 2: Mean DAG recovery performance, plus/minus one standard deviation, under heteroscedastic noise for both ER4 (top row) and SF4 (bottom row) graphs with varying numbers of nodes. Each column corresponds to a different noise distribution.

Table 2: DAG recovery results for 200-node ER4 graphs under heteroscedastic Gaussian noise.

|       | GOLEM-EV | GOLEM-NV | DAGMA | SortNRegress | CoLiDE-EV | CoLiDE-NV |
|-------|----------|----------|-------|--------------|-----------|-----------|
| SHD   | 642.9±61.5 | 481.3±45.8 | 470.2±50.6 | 397.8±27.8 | 426.5±49.6 | **390.7±35.6** |
| SID   | 29628±1008 | 25699±2194 | 24980±1456 | **22560±1749** | 23326±1620 | 22734±1767 |
| SHD-C | 630.1±53.5 | 509.5±50.6 | 490.2±46.8 | **402.1±26.3** | 449.4±44.9 | 407.9±37.5 |
| FDR   | 0.35±0.05 | 0.30±0.03 | 0.31±0.04 | **0.20±0.01** | 0.29±0.04 | 0.25±0.03 |
| TPR   | 0.33±0.09 | 0.60±0.07 | 0.64±0.02 | 0.62±0.02 | 0.68±0.02 | **0.68±0.01** |

This compressed interval, compared to the one used in the previous section, has a reduced signal-to-noise ratio (SNR) (Zheng et al., 2018; Reisach et al., 2021), which obfuscates the optimization.

Figure 2 presents experiments varying noise distributions, graph types, and node numbers. CoLiDE-NV is the clear winner, outperforming the alternatives in virtually all variations. Remarkably, CoLiDE-EV performs very well and often is the second-best solution, outmatching GOLEM-NV, even considering that an EV formulation is clearly underspecifying the problem.

These trends are confirmed in Table 2, where both CoLiDE formulations show strong efficacy on additional metrics (we point out that SHD is often considered the more accurate among all of them) for the ER4 instance with Gaussian noise. In this particular instance, SortNRegress is competitive with CoLiDE-NV in SID, SHD-C, and FDR. Note that, CoLiDE-NV consistently maintains lower deviations than DAGMA and GOLEM, underscoring its robustness. All in all, CoLiDE-NV is leading the pack in terms of SHD and performs very strongly across all other metrics.

In Appendix E.5, we include additional experiments where we draw the edge weights uniformly from $[-2, -0.5] \cup [0.5, 2]$. Here, CoLiDE-NV and CoLiDE-EV are evenly matched as the simplified problem complexity does not warrant the adoption of a more intricate formulation like CoLiDE-NV.

**Beyond continuous optimization.** We also tested the CoLiDE objective in combination with TOPO Deng et al. (2023a), a recently proposed bi-level optimization framework. Results in Appendix E.8 show that CoLiDE yields improvements when using this new optimization method.

**Noise estimation.** An accurate noise estimation is the crux of our work (Section 4), leading to a new formulation and algorithm for DAG learning. A method's ability to estimate noise variance reflects its proficiency in recovering accurate edge weights. Metrics such as SHD and TPR prioritize the detection of correct edges, irrespective of whether the edge weight closely matches the actual value. For methods that do not explicitly estimate the noise, we can evaluate them a posteriori by estimating the DAG first and subsequently computing the noise variances using the residual variance formula $\hat{\sigma}^2 = \frac{1}{dn}\|\mathbf{X} - \hat{\mathbf{W}}^\top\mathbf{X}\|_F^2$ in the EV case, or, $\hat{\sigma_i}^2 = \frac{1}{n}\|x_i - \hat{\mathbf{w}}_i^\top\mathbf{x}\|_2^2$ in the NV case.

We can now examine whether CoLiDE surpasses other state-of-the-art approaches in noise estimation. To this end, we generated 10 distinct ER4 DAGs with 200 nodes, employing the homo and heteroscedastic settings described earlier in this section. Assuming a Gaussian noise distribution, we

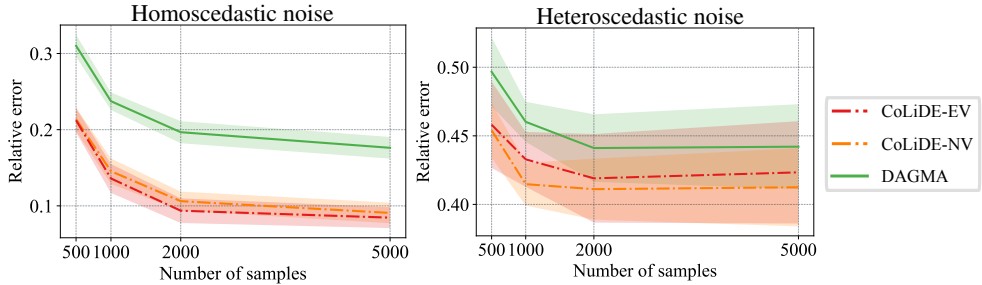

Figure 3: Mean relative noise estimation errors, plus/minus one standard deviation, as a function of the number of samples, aggregated from ten separate ER4 graphs, each comprising 200 nodes.

Table 3: DAG recovery performance on the Sachs dataset (Sachs et al., 2005).

|  | GOLEM-EV | GOLEM-NV | DAGMA | SortNRegress | DAGuerreotype | GES | CoLiDE-EV | CoLiDE-NV |
|---|---|---|---|---|---|---|---|---|
| SHD | 22 | 15 | 16 | 13 | 14 | 13 | 13 | **12** |
| SID | 49 | 58 | 52 | 47 | 50 | 56 | 47 | **46** |
| SHD-C | 19 | **11** | 15 | 13 | 12 | **11** | 13 | 14 |
| FDR | 0.83 | 0.66 | **0.5** | 0.61 | 0.57 | **0.5** | 0.54 | 0.53 |
| TPR | 0.11 | 0.11 | 0.05 | 0.29 | 0.17 | 0.23 | 0.29 | **0.35** |

generated different numbers of samples to assess CoLiDE's performance under limited data scenarios. We exclude GOLEM from the comparison due to its subpar performance compared to DAGMA. In the equal noise variance scenario, depicted in Figure 3 (left), CoLiDE-EV consistently outperforms DAGMA across varying sample sizes. CoLiDE-EV is also slightly better than CoLiDE-NV, due to its better modeling of the homoscedastic case. In Figure 3 (right), we examine the non-equal noise variance scenario, where CoLiDE-NV demonstrates superior performance over both CoLiDE-EV and DAGMA across different sample sizes. Of particular interest is the fact that CoLiDE-NV provides a lower error even when using half as many samples as DAGMA.

**Real data.** To conclude our analysis, we extend our investigation to a real-world example, the Sachs dataset (Sachs et al., 2005), which is used extensively throughout the probabilistic graphical models literature. The Sachs dataset encompasses cytometric measurements of protein and phospholipid constituents in human immune system cells (Sachs et al., 2005). This dataset comprises 11 nodes and 853 observation samples. The associated DAG is established through experimental methods as outlined by Sachs et al. (2005), and it enjoys validation from the biological research community. Notably, the ground truth DAG in this dataset consists of 17 edges. The outcomes of this experiment are consolidated in Table 3. Evidently, the results illustrate that CoLiDE-NV outperforms all other state-of-the-art methods, achieving a SHD of 12. To our knowledge, this represents the lowest achieved SHD among continuous optimization-based techniques applied to the Sachs problem.

## 6 CONCLUDING SUMMARY, LIMITATIONS, AND FUTURE WORK

In this paper we introduced CoLiDE, a framework for learning linear DAGs wherein we simultaneously estimate both the DAG structure and the exogenous noise levels. We present variants of CoLiDE to estimate homoscedastic and heteroscedastic noise across nodes. Additionally, estimating the noise eliminates the necessity for fine-tuning the model hyperparameters (e.g., the weight on the sparsity penalty) based on the *unknown* noise levels. Extensive experimental results have validated CoLiDE's superior performance when compared to other state-of-the-art methods in diverse synthetic and real-world settings, including the recovery of the DAG edges as well as their weights.

The scope of our DAG estimation framework is limited to observational data adhering to a linear SEM. In future work, we will extend it to encompass nonlinear and interventional settings. Therein, CoLiDE's formulation, amenable to first-order optimization, will facilitate a symbiosis with neural networks to parameterize SEM nonlinearities. Although CoLiDE's decomposability is a demonstrable property, further experimental results are needed to fully assert the practical value of stochastic optimization. Finally, we plan to introduce new optimization techniques to realize CoLiDE's full potential both in batch and online settings, with envisioned impact also to order-based methods.

## ACKNOWLEDGMENTS

This research was partially conducted while the first author was an intern at Intel Labs. The authors would like to thanks the anonymous reviewers for their thoughtful feedback and valuable suggestions on the original version of this manuscript, which led to a markedly improved revised paper.

## REPRODUCIBILITY STATEMENT

We have made our code publicly available at `https://github.com/SAMiatto/colide`. CoLiDE-related algorithmic details including hyperparameter selection are given in Section 4.1. Additional implementation details are in Appendix D.

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

# A  ADDITIONAL RELATED WORK

While the focus in the paper has been on continuous relaxation algorithms to tackle the linear DAG learning problem, numerous alternative *combinatorial search* approaches have been explored as well; see (Kitson et al., 2023) and (Vowels et al., 2022) for up-to-date tutorial expositions that also survey a host of approaches for *nonlinear SEMs*. For completeness, here we augment our Section 3 review of continuous relaxation and order-based methods to also account for discrete optimization alternatives in the literature.

**Discrete optimization methods.** A broad swath of approaches falls under the category of score-based methods, where (e.g., BD, BIC, BDe, MDL) scoring functions are used to guide the search for DAGs in $\mathbb{D}$, e.g., (Peters et al., 2017, Chapter 7.2.2). A subset of studies within this domain introduces modifications to the original problem by incorporating additional assumptions regarding the DAG or the number of parent nodes associated with each variable (Nie et al., 2014; Chen et al., 2016; Viinikka et al., 2020). Another category of discrete optimization methods is rooted in greedy search strategies or discrete optimization techniques applied to the determination of topological orders (Chickering, 2002; Park & Klabjan, 2017). GES (Ramsey et al., 2017) is a scalable greedy algorithm for discovering DAGs that we chose as one of our baselines. Constraint-based methods represent another broad category within discrete optimization approaches (Bühlmann et al., 2014). These approaches navigate $\mathbb{D}$ by conducting independence tests among observed variables; see e.g., (Peters et al., 2017, Chapter 7.2.1). Overall, many of these combinatorial search methods exhibit scalability issues, particularly when confronted with high-dimensional settings arising with large DAGs. This challenge arises because the space of possible DAGs grows at a superexponential rate with the number of nodes, e.g., Chickering (2002).

## A.1  BACKGROUND ON SMOOTHED CONCOMITANT LASSO ESTIMATORS

Consider a linear regression setting $\mathbf{y} = \mathbf{H}\boldsymbol{\theta} + \boldsymbol{\epsilon}$ in which we have access to a response vector $\mathbf{y} \in \mathbb{R}^d$ and a design matrix $\mathbf{H} \in \mathbb{R}^{d \times p}$ comprising $p$ explanatory variables or features. To obtain a sparse vector of regression coefficients $\boldsymbol{\theta} \in \mathbb{R}^p$, one can utilize the convex lasso estimator (Tibshirani, 1996)

$$\hat{\boldsymbol{\theta}} \in \operatorname*{argmin}_{\boldsymbol{\theta}} \frac{1}{2d}\|\mathbf{y} - \mathbf{H}\boldsymbol{\theta}\|_2^2 + \lambda\|\boldsymbol{\theta}\|_1 \tag{7}$$

which facilitates continuous estimation and variable selection. Statistical guarantees for lasso hinge on selecting the scalar parameter $\lambda > 0$ to be proportional to the noise level. In particular, under the assumption that $\boldsymbol{\epsilon} \sim \mathcal{N}(0, \sigma^2 \mathbf{I}_d)$, solving (7) with $\lambda^* \asymp \sigma\sqrt{\frac{\log p}{d}}$ yields the minimax optimal solution for parameter estimation in high dimensions (Bickel et al., 2009; Li et al., 2020). However, in most cases having knowledge of the noise variance is a luxury we may not possess.

To address the aforementioned challenge, a promising solution involves the simultaneous estimation of both sparse regression coefficients and the noise level. Over the years, various formulations have been proposed to tackle this problem, ranging from penalized maximum likelihood approaches (Städler et al., 2010) to frameworks inspired by robust theory (Huber, 1981; Owen, 2007). Several of these methods are closely related; in particular, the concomitant lasso approach in (Owen, 2007) has been shown to be equivalent to the so-termed square-root lasso (Belloni et al., 2011), and rediscovered as the scaled lasso estimator (Sun & Zhang, 2012). Notably, the *smoothed* concomitant lasso (Ndiaye et al., 2017) stands out as the most recent and efficient method suitable for high-dimensional settings. The approach in (Ndiaye et al., 2017) is to jointly estimate the regression coefficients $\boldsymbol{\theta}$ and the noise level $\sigma$ by solving the *jointly convex* problem

$$\min_{\boldsymbol{\theta}, \sigma} \frac{1}{2d\sigma}\|\mathbf{y} - \mathbf{H}\boldsymbol{\theta}\|_2^2 + \frac{\sigma}{2} + \lambda\|\boldsymbol{\theta}\|_1 + \mathbb{I}\{\sigma \geq \sigma_0\}, \tag{8}$$

where $\sigma_0$ is a predetermined lower bound based on either prior information or proportional to the initial noise standard deviation, e.g., $\sigma_0 = \frac{\|\mathbf{y}\|_2}{\sqrt{d}} \times 10^{-2}$. Inclusion of the constraint $\sigma \geq \sigma_0$ is motivated in (Ndiaye et al., 2017) to prevent ill-conditioning as the solution $\hat{\sigma}$ approaches zero. To solve (8), a BCD algorithm is adopted wherein one iteratively (and cyclically) solves (8) for $\boldsymbol{\theta}$ with fixed $\sigma$, and subsequently updates the value of $\sigma$ using a closed-form solution given the most up-to-date value of $\boldsymbol{\theta}$; see (Ndiaye et al., 2017) for further details.

Interestingly, disregarding the constraint $\sigma \geq \sigma_0$ and plugging the noise estimator $\hat{\sigma} = \|\mathbf{y} - \mathbf{H}\boldsymbol{\theta}\|_2/\sqrt{d}$ in (8), yields the squared-root lasso problem (Belloni et al., 2011), i.e.,

$$\min_{\boldsymbol{\theta}} \frac{1}{\sqrt{d}}\|\mathbf{y} - \mathbf{H}\boldsymbol{\theta}\|_2 + \lambda\|\boldsymbol{\theta}\|_1. \tag{9}$$

This problem is convex but with the added difficulty of being non-smooth when $\mathbf{y} = \mathbf{H}\boldsymbol{\theta}$. It has been proven that the solutions to problems (8) and (9) are equivalent (Ndiaye et al., 2017). Moreover, one can show that if $\boldsymbol{\epsilon} \sim \mathcal{N}(0, \sigma^2\mathbf{I}_d)$ and $\lambda^* \asymp \sqrt{\frac{\log p}{d}}$, the solution of (9) is minimax optimal, and notably, it is independent of the noise scale $\sigma$.

**Remark.** Li et al. (2020) noticed that the non-differentiability of the squared-root lasso is not an issue, in the sense that a subgradient can be used safely, if one is guaranteed to avoid the singularity. For DAG estimation, due to the exogenous noise in the linear SEM, we are exactly in this situation. However, we point out that this alternative makes the objective function not separable across samples, precluding stochastic optimization that could be desirable for scalability. Nonetheless, we leave the exploration of this alternative as future work.

A limitation of the smoothed concomitant lasso in (8) is its inability to handle *multi-task* settings where response data $\mathbf{Y} \in \mathbb{R}^{d \times n}$ is collected from $d$ diverse sources with varying noise levels. Statistical and algorithmic issues pertaining to this heteroscedastic scenario have been successfully addressed in (Massias et al., 2018), by generalizing the smoothed concomitant lasso formulation to perform joint estimation of the coefficient matrix $\boldsymbol{\Theta} \in \mathbb{R}^{p \times n}$ and the *square-root* of the noise covariance matrix $\boldsymbol{\Sigma} = \text{diag}(\sigma_1, \ldots, \sigma_d) \in \mathbb{R}^{d \times d}$. Accordingly, the so-termed generalized concomitant multi-task lasso is given by

$$\min_{\boldsymbol{\Theta}, \boldsymbol{\Sigma}} \frac{1}{2nd}\|\mathbf{Y} - \mathbf{H}\boldsymbol{\Theta}\|_{\boldsymbol{\Sigma}^{-1}}^2 + \frac{\text{tr}(\boldsymbol{\Sigma})}{2d} + \lambda\|\boldsymbol{\Theta}\|_{2,1} + \mathbb{I}\{\boldsymbol{\Sigma} \geq \boldsymbol{\Sigma}_0\}. \tag{10}$$

Similar to the smoothed concomitant lasso, the matrix $\boldsymbol{\Sigma}_0$ is either assumed to be known a priori, or, it is estimated from the data, e.g., via $\boldsymbol{\Sigma}_0 = \frac{\|\mathbf{Y}\|_F}{\sqrt{nd}} \times 10^{-2}$ as suggested in (Massias et al., 2018).

## B   ALGORITHMIC DERIVATIONS

In this section, we compute the gradients of the proposed score functions with respect to $\mathbf{W}$, derive closed-form solutions for updating $\sigma$ and $\boldsymbol{\Sigma}$, and discuss the associated computational complexity.

### B.1   EQUAL NOISE VARIANCE

**Gradients.** We introduced the score function $\mathcal{S}(\mathbf{W}, \sigma) = \frac{1}{2n\sigma}\|\mathbf{X} - \mathbf{W}^\top\mathbf{X}\|_F^2 + \frac{d\sigma}{2} + \lambda\|\mathbf{W}\|_1$ in (2). Defining $f(\mathbf{W}, \sigma) := \frac{1}{2n\sigma}\|\mathbf{X} - \mathbf{W}^\top\mathbf{X}\|_F^2 + \frac{d\sigma}{2}$, we calculate the gradient of $f(\mathbf{W}, \sigma)$ w.r.t. $\mathbf{W}$, for fixed $\sigma = \hat{\sigma}$.

The smooth terms in the score function can be rewritten as

$$f(\mathbf{W}, \hat{\sigma}) = \frac{1}{2n\hat{\sigma}}\|\mathbf{X} - \mathbf{W}^\top\mathbf{X}\|_F^2 + \frac{d\hat{\sigma}}{2} \tag{11}$$

$$= \frac{1}{2n\hat{\sigma}}\text{Tr}\left((\mathbf{X} - \mathbf{W}^\top\mathbf{X})^\top(\mathbf{X} - \mathbf{W}^\top\mathbf{X})\right) + \frac{d\hat{\sigma}}{2} \tag{12}$$

$$= \frac{1}{2n\hat{\sigma}}\text{Tr}\left(\mathbf{X}^\top(\mathbf{I} - \mathbf{W})(\mathbf{I} - \mathbf{W}^\top)\mathbf{X}\right) + \frac{d\hat{\sigma}}{2} \tag{13}$$

$$= \frac{1}{2\hat{\sigma}}\text{Tr}\left((\mathbf{I} - \mathbf{W})^\top\text{cov}(\mathbf{X})(\mathbf{I} - \mathbf{W})\right) + \frac{d\hat{\sigma}}{2}, \tag{14}$$

where $\text{cov}(\mathbf{X}) = \frac{1}{n}\mathbf{X}\mathbf{X}^\top$ is the sample covariance matrix. Accordingly, the gradient of $f(\mathbf{W}, \sigma)$ is

$$\nabla_{\mathbf{W}} f(\mathbf{W}, \hat{\sigma}) = -\frac{(\mathbf{X}\mathbf{X}^\top)}{n\hat{\sigma}} + \frac{(\mathbf{X}\mathbf{X}^\top)\mathbf{W}}{n\hat{\sigma}} \tag{15}$$

$$= -\frac{1}{\hat{\sigma}}\text{cov}(\mathbf{X})[\mathbf{I} - \mathbf{W}]. \tag{16}$$

**Closed-form solution for $\hat{\sigma}$ in** (4). We update $\sigma$ by fixing $\mathbf{W}$ to $\hat{\mathbf{W}}$ and computing the minimizer in closed form. The first-order optimality condition yields

$$\frac{\partial f(\hat{\mathbf{W}}, \sigma)}{\partial \sigma} = \frac{-1}{2n\sigma^2} \|\mathbf{X} - \hat{\mathbf{W}}^\top \mathbf{X}\|_F^2 + \frac{d}{2} = 0. \tag{17}$$

Hence,

$$\hat{\sigma}^2 = \frac{1}{nd} \|\mathbf{X} - \hat{\mathbf{W}}^\top \mathbf{X}\|_F^2 \tag{18}$$

$$= \frac{1}{nd} \operatorname{Tr}\left((\mathbf{X} - \hat{\mathbf{W}}^\top \mathbf{X})^\top (\mathbf{X} - \hat{\mathbf{W}}^\top \mathbf{X})\right) \tag{19}$$

$$= \frac{1}{nd} \operatorname{Tr}\left(\mathbf{X}^\top (\mathbf{I} - \hat{\mathbf{W}})(\mathbf{I} - \hat{\mathbf{W}}^\top)\mathbf{X}\right) \tag{20}$$

$$= \frac{1}{nd} \operatorname{Tr}\left((\mathbf{I} - \hat{\mathbf{W}}^\top)\mathbf{X}\mathbf{X}^\top (\mathbf{I} - \hat{\mathbf{W}})\right) \tag{21}$$

$$= \frac{1}{d} \operatorname{Tr}\left((\mathbf{I} - \hat{\mathbf{W}})^\top \operatorname{cov}(\mathbf{X})(\mathbf{I} - \hat{\mathbf{W}})\right). \tag{22}$$

Because of the constraint $\sigma \geq \sigma_0$, the minimizer is given by

$$\hat{\sigma} = \max\left(\sqrt{\operatorname{Tr}\left((\mathbf{I} - \mathbf{W})^\top \operatorname{cov}(\mathbf{X})(\mathbf{I} - \mathbf{W})\right)/d}, \sigma_0\right). \tag{23}$$

**Complexity.** The gradient with respect to $\mathbf{W}$ involves subtracting two matrices of size $d \times d$, resulting in a computational complexity of $\mathcal{O}(d^2)$. Additionally, three matrix multiplications contribute to a complexity of $\mathcal{O}(nd^2 + d^3)$. The subsequent element-wise division adds $\mathcal{O}(d^2)$. Therefore, the main computational complexity of gradient computation is $\mathcal{O}(d^3)$, on par with state-of-the-art continuous optimization methods for DAG learning. Similarly, the closed-form solution for updating $\sigma$ has a computational complexity of $\mathcal{O}(d^3)$, involving two matrix subtractions ($\mathcal{O}(2d^2)$), four matrix multiplications ($\mathcal{O}(nd^2 + 2d^3)$), and matrix trace operations ($\mathcal{O}(d)$).

**Online updates.** The closed-form solution for $\hat{\sigma}$ in (4) can be used in a batch setting when all samples are jointly available. The proposed score function allows to design a solution for this scenario, as the updates naturally decompose in time (i.e., across samples). In the mini-batch setting, at each iteration $t$, we have access to a randomly selected subset of data $\mathbf{X}_t \in \mathbb{R}^{d \times n_b}$ with $n_b < n$ as the size of the mini-batch. Consequently, we could utilize mini-batch stochastic gradient descent as the optimization method within our framework. Given an initial solution $\hat{\mathbf{W}}_0$, it is possible to conceptualize an online algorithm with the following iterations for $t = 1, 2, \ldots$

1. Compute the sample covariance matrix $\operatorname{cov}(\mathbf{X}_t) = \frac{1}{t}\left((t-1)\operatorname{cov}(\mathbf{X}_{t-1}) + \frac{1}{n_b}\mathbf{X}_t\mathbf{X}_t^\top\right)$.
2. Compute $\hat{\mathbf{W}}_t$ using $\operatorname{cov}(\mathbf{X}_t)$ and a first-order update as in Algorithm 1.
3. Compute the noise estimate $\hat{\sigma}_t$ using $\operatorname{cov}(\mathbf{X}_t)$ and (4).

Alternatively, we could keep sufficient statistics for the noise estimate. In this case, the update would proceed as follows. We first compute the residual $\epsilon_t = \frac{1}{n_b d}\|\mathbf{X}_t - \hat{\mathbf{W}}_{t-1}^\top \mathbf{X}_t\|_2^2$ and update the sufficient statistic $e_t = e_{t-1} + \epsilon_t$. Finally, we compute the noise estimate $\hat{\sigma}_t = \max\left(\sqrt{\frac{1}{t}e_t}, \sigma_0\right)$. The use of this type of sufficient statistics in online algorithms has a long-standing tradition in the adaptive signal processing and online learning literature (e.g., Mairal et al., 2010). Although the above iterations form the conceptual sketch for an online algorithm, many important details need to be addressed to achieve a full online solution and we leave this work for the future. Our goal here was to illustrate that resulting DAG structure and scale estimators are decomposable across samples.

To empirically explore the mentioned sufficient statistics, we conducted an experiment using mini-batches of data. The performance of these optimization methods is tested on an ER4 graph of size $d = 50$, where we generate $n = 1000$ i.i.d samples using a linear SEM. We assume the noise distribution is Gaussian, and the noise variances are equal. We explore two distinct batch sizes, namely 100 and 500, corresponding to using 10% and 50% of the data at each iteration, respectively. To monitor how well these algorithms track the output of the CoLiDE-EV algorithm, denoted as $\mathbf{W}^\star$ and $\sigma^\star$, we calculate the relative error $\frac{\|\mathbf{W}_t - \mathbf{W}^\star\|_F}{\|\mathbf{W}^\star\|_F}$ at each iteration $t$. Similarly, the relative error

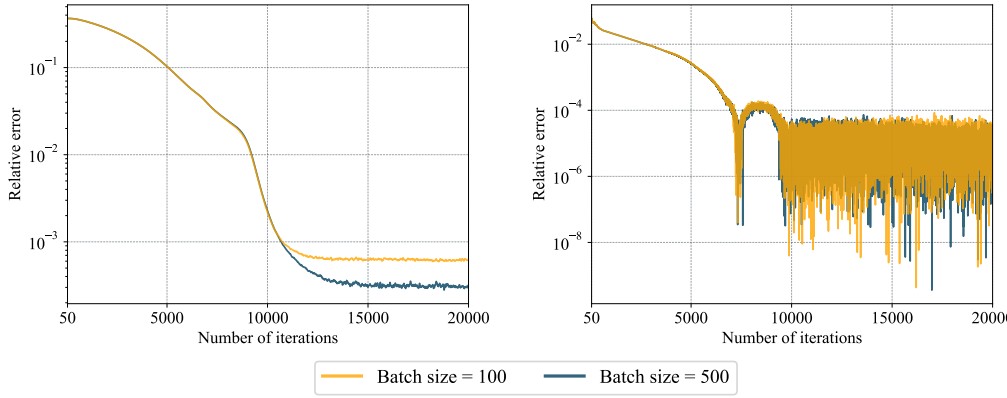

Figure 4: Tracking performance of mini-batch stochastic gradient descent in relation to the output of the original CoLiDE-EV algorithm. The left plot illustrates the tracking of the output graph $\mathbf{W}^\star$, while the right plot represents the tracking of the noise level $\sigma^\star$.

for the noise scale is computed as $\frac{|\sigma_t - \sigma^\star|}{|\sigma^\star|}$. As outlined in Section 4, we address CoLiDE-EV for sequences of decreasing $\mu_k$. To enhance visual clarity, we focus on $\mathbf{W}_t$ and $\sigma_t$ for the smallest $\mu_k$.

The experimental results, shown in Figure 4, demonstrate that mini-batch stochastic gradient descent with varying batch sizes adeptly follows the output of CoLiDE-EV for both the DAG adjacency matrix (left) and the noise level (right). While these findings showcase promising results, we acknowledge the need for further evaluation to conclusively assert the decomposability benefits of our method. This experiment serves as an initial exploration, providing valuable insights for a subsequent analysis of the online updates.

## B.2 NON-EQUAL NOISE VARIANCE

**Gradients.** We introduced the score function $\mathcal{S}(\mathbf{W}, \boldsymbol{\Sigma})$ in (5). Upon defining

$$g(\mathbf{W}, \boldsymbol{\Sigma}) = \frac{1}{2n} \operatorname{Tr}\left((\mathbf{X} - \mathbf{W}^\top \mathbf{X})^\top \boldsymbol{\Sigma}^{-1} (\mathbf{X} - \mathbf{W}^\top \mathbf{X})\right) + \frac{1}{2} \operatorname{Tr}(\boldsymbol{\Sigma}), \tag{24}$$

we derive the gradient of $g(\mathbf{W}, \boldsymbol{\Sigma})$ w.r.t. $\mathbf{W}$, while maintaining $\boldsymbol{\Sigma} = \hat{\boldsymbol{\Sigma}}$ fixed. To this end, note that $g(\mathbf{W}, \hat{\boldsymbol{\Sigma}})$ can be rewritten as

$$g(\mathbf{W}, \hat{\boldsymbol{\Sigma}}) = \frac{1}{2n} \operatorname{Tr}\left((\mathbf{X} - \mathbf{W}^\top \mathbf{X})^\top \hat{\boldsymbol{\Sigma}}^{-1} (\mathbf{X} - \mathbf{W}^\top \mathbf{X})\right) + \frac{1}{2} \operatorname{Tr}(\hat{\boldsymbol{\Sigma}}) \tag{25}$$

$$= \frac{1}{2n} \operatorname{Tr}\left(\hat{\boldsymbol{\Sigma}}^{-1} (\mathbf{X} - \mathbf{W}^\top \mathbf{X})(\mathbf{X} - \mathbf{W}^\top \mathbf{X})^\top\right) + \frac{1}{2} \operatorname{Tr}(\hat{\boldsymbol{\Sigma}}) \tag{26}$$

$$= \frac{1}{2} \operatorname{Tr}\left(\hat{\boldsymbol{\Sigma}}^{-1} (\mathbf{I} - \mathbf{W}^\top) \frac{\mathbf{X}\mathbf{X}^\top}{n} (\mathbf{I} - \mathbf{W})\right) + \frac{1}{2} \operatorname{Tr}(\hat{\boldsymbol{\Sigma}}) \tag{27}$$

$$= \frac{1}{2} \operatorname{Tr}\left(\hat{\boldsymbol{\Sigma}}^{-1} (\mathbf{I} - \mathbf{W})^\top \operatorname{cov}(\mathbf{X})(\mathbf{I} - \mathbf{W})\right) + \frac{1}{2} \operatorname{Tr}(\hat{\boldsymbol{\Sigma}}). \tag{28}$$

Accordingly, the gradient of $g(\mathbf{W}, \boldsymbol{\Sigma})$ is

$$\nabla_{\mathbf{W}} g(\mathbf{W}, \hat{\boldsymbol{\Sigma}}) = \frac{1}{2n} \left[-\mathbf{X}\mathbf{X}^\top \hat{\boldsymbol{\Sigma}}^{-\top} - \mathbf{X}\mathbf{X}^\top \hat{\boldsymbol{\Sigma}}^{-1} + \mathbf{X}\mathbf{X}^\top \mathbf{W} \hat{\boldsymbol{\Sigma}}^{-\top} + \mathbf{X}\mathbf{X}^\top \mathbf{W} \hat{\boldsymbol{\Sigma}}^{-1}\right] \tag{29}$$

$$= \frac{-\mathbf{X}\mathbf{X}^\top \hat{\boldsymbol{\Sigma}}^{-1}}{n} + \frac{\mathbf{X}\mathbf{X}^\top \mathbf{W} \hat{\boldsymbol{\Sigma}}^{-1}}{n} \tag{30}$$

$$= -\operatorname{cov}(\mathbf{X}) \left[\mathbf{I} - \mathbf{W}\right] \hat{\boldsymbol{\Sigma}}^{-1}. \tag{31}$$

**Closed-form solution for $\hat{\boldsymbol{\Sigma}}$ in** (6). We update $\boldsymbol{\Sigma}$ by fixing $\mathbf{W}$ to $\hat{\mathbf{W}}$ and computing the minimizer in closed form. The first-order optimality condition yields

$$\nabla_{\boldsymbol{\Sigma}} g(\hat{\mathbf{W}}, \boldsymbol{\Sigma}) = \frac{-\boldsymbol{\Sigma}^{-\top}}{2n}(\mathbf{X} - \hat{\mathbf{W}}^{\top}\mathbf{X})(\mathbf{X} - \hat{\mathbf{W}}^{\top}\mathbf{X})^{\top}\boldsymbol{\Sigma}^{-\top} + \frac{1}{2}\mathbf{I} = \mathbf{0}. \tag{32}$$

Hence,

$$\hat{\boldsymbol{\Sigma}}^2 = \frac{(\mathbf{X} - \hat{\mathbf{W}}^{\top}\mathbf{X})(\mathbf{X} - \hat{\mathbf{W}}^{\top}\mathbf{X})^{\top}}{n} \tag{33}$$

$$= (\mathbf{I} - \hat{\mathbf{W}})^{\top} \operatorname{cov}(\mathbf{X})(\mathbf{I} - \hat{\mathbf{W}}). \tag{34}$$

Note that we can ascertain $\hat{\boldsymbol{\Sigma}}$ is a diagonal matrix, based on SEM assumptions of exogenous noises being mutually independent. Because of the constraint $\boldsymbol{\Sigma} \geq \boldsymbol{\Sigma}_0$, the minimizer is given by

$$\hat{\boldsymbol{\Sigma}} = \max\left(\sqrt{\operatorname{diag}\left((\mathbf{I} - \mathbf{W})^{\top} \operatorname{cov}(\mathbf{X})(\mathbf{I} - \mathbf{W})\right)}, \boldsymbol{\Sigma}_0\right), \tag{35}$$

where $\sqrt{(\cdot)}$ and $\max(\cdot)$ indicates an element-wise operation, while the operator $\operatorname{diag}(\cdot)$ extracts the diagonal elements of a matrix.

**Complexity.** The gradient with respect to $\mathbf{W}$ entails the subtraction of two $d \times d$ matrices, resulting in $\mathcal{O}(d^2)$ complexity. The inverse of a diagonal matrix, involved in the calculation, also incurs a complexity of $\mathcal{O}(d^2)$. Additionally, four matrix multiplications contribute to a complexity of $\mathcal{O}(nd^2 + 2d^3)$. The subsequent element-wise division introduces an additional $\mathcal{O}(d^2)$. Consequently, the principal computational complexity of gradient computation is $\mathcal{O}(d^3)$, aligning with the computational complexity of state-of-the-art methods. The closed-form solution for updating $\boldsymbol{\Sigma}$ has a computational complexity of $\mathcal{O}(d^3)$, involving two matrix subtractions ($\mathcal{O}(2d^2)$), four matrix multiplications ($\mathcal{O}(nd^2 + 2d^3)$), and two element-wise operations on the diagonal elements of a $d \times d$ matrix ($\mathcal{O}(2d)$).

**Online updates.** As in the homoscedastic case, we can devise an online algorithm based on CoLiDE-NV. The main difference is that we now need a collection of $d$ sufficient statistics, one for each node in the graph. For $j = 1, \ldots, d$ and all $t = 1, 2, \ldots$, we now have

$$\epsilon_{j,t} = \left((\mathbf{x}_t)_j - (\hat{\mathbf{W}}_{t-1}^{\top}\mathbf{x}_t)_j\right)^2, \tag{36}$$

$$e_{j,t} = e_{j,t-1} + \epsilon_{j,t}, \tag{37}$$

$$\hat{\sigma}_{j,t} = \max\left(\sqrt{\frac{1}{t}e_{j,t}}, \sigma_0\right), \tag{38}$$

where $\mathbf{x}_t = [(\mathbf{x}_t)_1, \ldots, (\mathbf{x}_t)_d]$ and $e_{j,0} = 0$ for all $j$. Using $\mathbf{x}_t$, the noise estimates $\{\hat{\sigma}_{1,t}, \ldots, \hat{\sigma}_{d,t}\}$, and $\hat{\mathbf{W}}_{t-1}$, we can compute $\hat{\mathbf{W}}_t$, e.g., using a first-order update as in Algorithm 1. Although these elements provide the foundation for an online algorithm, we leave the completion of this new technique as future work. In any case, this shows the updates decompose across samples.

### B.3 GRADIENT OF THE LOG-DETERMINANT ACYCLICITY FUNCTION

We adopt $\mathcal{H}_{\mathrm{ldet}}(\mathbf{W}, s) = d\log(s) - \log(\det(s\mathbf{I} - \mathbf{W} \circ \mathbf{W}))$ as the acyclicity function in CoLiDE's formulation. As reported in Bello et al. (2022), the gradient of $\mathcal{H}_{\mathrm{ldet}}(\mathbf{W}, s)$ is given by

$$\nabla\mathcal{H}_{\mathrm{ldet}}(\mathbf{W}) = 2\left[s\mathbf{I} - \mathbf{W} \circ \mathbf{W}\right]^{-\top} \circ \mathbf{W}. \tag{39}$$

The computational complexity incurred in each gradient evaluation is $\mathcal{O}(d^3)$. This includes a matrix subtraction ($\mathcal{O}(d^2)$), four element-wise operations ($\mathcal{O}(4d^2)$), and a matrix inversion ($\mathcal{O}(d^3)$).

**Complexity summary.** All in all, both CoLiDE variants, CoLiDE-EV and CoLiDE-NV, incur a per iteration computational complexity of $\mathcal{O}(d^3)$. While CoLiDE concurrently estimates the unknown noise levels along with the DAG topology, this comes with no significant computational complexity overhead relative to state-of-the-art continuous relaxation methods for DAG learning.

## C  GUARANTEES FOR (HETEROSCEDASTIC) LINEAR GAUSSIAN SEMS

Consider a general linear Gaussian SEM, whose exogenous noises can have non-equal variances (i.e., the heteroscedastic case). This is a non-identifiable model (Peters et al., 2017), meaning that the ground-truth DAG cannot be uniquely recovered from observational data alone. Still, we argue that the interesting theoretical analysis framework put forth in (Ghassami et al., 2020) and (Ng et al., 2020) – as well as its conclusions – carry over to CoLiDE. The upshot is that just like GOLEM, the solution of the CoLiDE-NV problem with $\mu_k \to \infty$ asymptotically (in $n$) will be a DAG *equivalent* to the ground truth. The precise notion of (quasi)-equivalence among directed graphs was introduced by Ghassami et al. (2020), and we will not reproduce all definitions and technical details here. The interested reader is referred to (Ng et al., 2020, Section 3.1).

Specifically, it follows that under the same assumptions in (Ng et al., 2020, Section 3.1), Theorems 1 and 2 therein hold for CoLiDE-NV when $\mu_k \to \infty$. Moreover, (Ng et al., 2020, Corollary 1) holds as well. This corollary motivates augmenting the score function in (5) with the DAG penalty $\mathcal{H}_{\mathrm{ldet}}(\mathbf{W}, s_k)$, to obtain a DAG solution quasi-equivalent to the ground truth DAG in lieu of the so-termed 'Triangle assumption'. The proofs of these results are essentially identical to the ones in (Ng et al., 2020, Appendix B), so in the sequel we only highlight the minor differences.

Let $\mathcal{G}^\star$ and $\boldsymbol{\Theta}$ be the ground truth DAG and the precision matrix of the random vector $\mathbf{x} \in \mathbb{R}^d$, so that the generated distribution is $\mathbf{x} \sim \mathcal{N}(\mathbf{0}, \boldsymbol{\Theta}^{-1})$. Let $\mathbf{W}$ and $\boldsymbol{\Sigma}^2$ be the weighted adjacency matrix and the diagonal matrix containing exogenous noise variances $\sigma_i^2$, respectively. As $n \to \infty$, the law of large numbers asserts that the sample covariance matrix $\mathrm{cov}(\mathbf{X}) \to \boldsymbol{\Theta}^{-1}$, almost surely. Then, if we drop the penalty term, $\mu_k \to \infty$, and considering $\lambda$ such that both trace terms in the score function in (5) dominate asymptotically, CoLiDE-NV's optimality condition implies [cf. (34)]

$$\hat{\boldsymbol{\Sigma}}^2 = (\mathbf{I} - \hat{\mathbf{W}})^\top \boldsymbol{\Theta}^{-1}(\mathbf{I} - \hat{\mathbf{W}}). \tag{40}$$

Note that when $\mu_k \to \infty$, CoLiDE-NV is convex and we are thus ensured to attain global optimality. This means that we will find a pair $(\hat{\mathbf{W}}, \hat{\boldsymbol{\Sigma}})$ of estimates such that $(\mathbf{I} - \hat{\mathbf{W}})\hat{\boldsymbol{\Sigma}}^{-2}(\mathbf{I} - \hat{\mathbf{W}})^\top = \boldsymbol{\Theta}$, and denote the directed graph corresponding to $\hat{\mathbf{W}}$ by $\hat{\mathcal{G}}(\hat{\mathbf{W}})$. One can readily show that under the mild assumptions in (Ng et al., 2020), $\hat{\mathcal{G}}(\hat{\mathbf{W}})$ is quasi equivalent to $\mathcal{G}^\star$. The remaining steps of the proofs of both Theorems are identical to those laid out by Ng et al. (2020, Appendix B).

## D  IMPLEMENTATION DETAILS

In this section, we provide a comprehensive description of the implementation details for the experiments conducted to evaluate and benchmark the proposed DAG learning algorithms.

### D.1  COMPUTING INFRASTRUCTURE

All experiments were executed on a 2-core Intel Xeon processor E5-2695v2 with a clock speed of 2.40 GHz and 32GB of RAM. For models like GOLEM and DAGuerreotype that necessitate GPU processing, we utilized either the NVIDIA A100, Tesla V100, or Tesla T4 GPUs.

### D.2  GRAPH MODELS

In our experiments, each simulation involves sampling a graph from two prominent random graph models:

- **Erdős-Rényi (ER):** These random graphs have independently added edges with equal probability. The chosen probability is determined by the desired nodal degree. Since ER graphs are undirected, we randomly generate a permutation vector for node ordering and orient the edges accordingly.

- **Scale-Free (SF):** These random graphs are generated using the preferential attachment process (Barabási & Albert, 1999). The number of edges preferentially attached is based on the desired nodal degree. The edges are oriented each time a new node is attached, resulting in a sampled directed acyclic graph (DAG).

### D.3 METRICS

We employ the following standard metrics commonly used in the context of DAG learning:

- **Structural Hamming distance (SHD):** Quantifies the total count of edge additions, deletions, and reversals required to transform the estimated graph into the true graph. Normalized SHD is obtained by dividing this count by the number of nodes.

- **SHD-C:** Initially, we map both the estimated graph and the ground truth DAG to their corresponding Completed Partially Directed Acyclic Graphs (CPDAGs). A CPDAG represents a Markov equivalence class encompassing all DAGs that encode identical conditional independencies. Subsequently, we calculate the SHD between the two CPDAGs. The normalized version of SHD-C is obtained by dividing the original metric by the number of nodes.

- **Structural Intervention Distance (SID):** Counts the number of causal paths that are disrupted in the predicted DAG. When divided by the number of nodes, we obtain the normalized SID.

- **True Positive Rate (TPR):** Measures the proportion of correctly identified edges relative to the total number of edges in the ground-truth DAG.

- **False Discovery Rate (FDR):** Represents the ratio of incorrectly identified edges to the total number of detected edges.

For all metrics except TPR, a *lower* value indicates better performance.

### D.4 NOISE DISTRIBUTIONS

We generate data by sampling from a set of linear SEMs, considering three distinct noise distributions:

- **Gaussian:** $z_i \sim \mathcal{N}(0, \sigma_i^2)$, $i = 1, \dots, d$, where the noise variance of each node is $\sigma_i^2$

- **Exponential:** $z_i \sim \text{Exp}(\lambda_i)$, $i = 1, \dots, d$, where the noise variance of each node is $\lambda_i^{-2}$.

- **Laplace:** $z_i \sim \text{Laplace}(0, b_i)$, $i = 1, \dots, d$, where the noise variance of each node is $2b_i^2$.

### D.5 BASELINE METHODS

To assess the performance of our proposed approach, we benchmark it against several state-of-the-art methods commonly recognized as baselines in this domain. All the methods are implemented using Python.

- **GOLEM:** This likelihood-based method was introduced by Ng et al. (2020). The code for GOLEM is publicly available on GitHub at `https://github.com/ignavier/golem`. Utilizing Tensorflow, GOLEM requires GPU support. We adopt their recommended hyperparameters for GOLEM. For GOLEM-EV, we set $\lambda_1 = 2 \times 10^{-2}$ and $\lambda_2 = 5$, and for GOLEM-NV, $\lambda_1 = 2 \times 10^{-3}$ and $\lambda_2 = 5$; refer to Appendix F of Ng et al. (2020) for details. Closely following the guidelines in Ng et al. (2020), we initialize GOLEM-NV with the output of GOLEM-EV.

- **DAGMA:** We employ linear DAGMA as introduced in Bello et al. (2022). The code is available at `https://github.com/kevinsbello/dagma`. We adhere to their recommended hyperparameters: $T = 4$, $s = \{1, .9, .8, .7\}$, $\beta_1 = 0.05$, $\alpha = 0.1$, and $\mu^{(0)} = 1$; consult Appendix C of Bello et al. (2022) for specifics.

- **GES:** This method utilizes a fast greedy approach for learning DAGs, outlined in Ramsey et al. (2017). The implementation leverages the `py-causal` Python package and can be accessed at `https://github.com/bd2kccd/py-causal`. We configure the hyperparameters as follows: `scoreId = 'cg-bic-score'`, `maxDegree = 5`, `dataType = 'continuous'`, and `faithfulnessAssumed = False`.

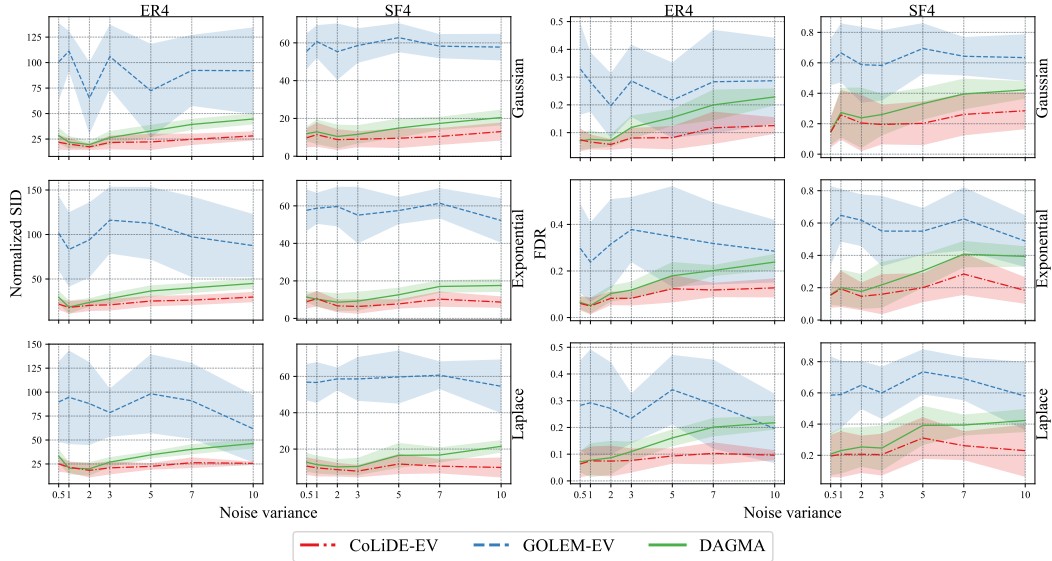

Figure 5: DAG recovery performance assessed for ER4 and SF4 graphs with 200 nodes, assuming equal noise variances. Each row represents a distinct noise distribution, and the shaded area depicts the standard deviation. The first two columns display SID, while the remaining columns focus on FDR.

- **SortNRegress:** This method adopts a two-step framework involving node ordering based on increasing variance and parent selection using the Least Angle Regressor (Reisach et al., 2021). The code is publicly available at `https://github.com/Scriddie/Varsortability`.

- **DAGuerreotype:** This recent approach employs a two-step framework to learn node orderings through permutation matrices and edge representations, either jointly or in a bilevel fashion (Zantedeschi et al., 2023). The implementation is accessible at `https://github.com/vzantedeschi/DAGuerreotype` and can utilize GPU processing. We employ the linear version of DAGuerreotype with `sparseMAP` as the operator for node ordering learning. Remaining parameters are set to defaults as per their paper. For real datasets, we use bi-level optimization with the following hyperparameters: $K = 100$, `pruning_reg=0.01`, `lr_theta=0.1`, and `standardize=True`. For synthetic simulations, we use joint optimization with $K = 10$.

## E ADDITIONAL EXPERIMENTS

Here, we present supplementary experimental results that were excluded from the main body of the paper due to page limitations.

### E.1 ADDITIONAL METRICS FOR THE HOMOSCEDASTIC EXPERIMENTS

For the homoscedastic experiments in Section 5, we exclusively presented the normalized SHD for 200-node graphs at various noise levels. To offer a comprehensive analysis, we now introduce additional metrics—SID and FDR—in Figure 5. We have also incorporated normalized SHD-C, as illustrated in Figure 6. The results in Figures 5 and 6 demonstrate the consistent superiority of CoLiDE-EV over other methods in both metrics, across all noise variances.

### E.2 SMALLER GRAPHS WITH HOMOSCEDASTIC ASSUMPTION

For the homoscedastic experiments in Section 5, we exclusively examined graphs comprising 200 nodes. Here, for a more comprehensive analysis, we extend our investigation to encompass smaller

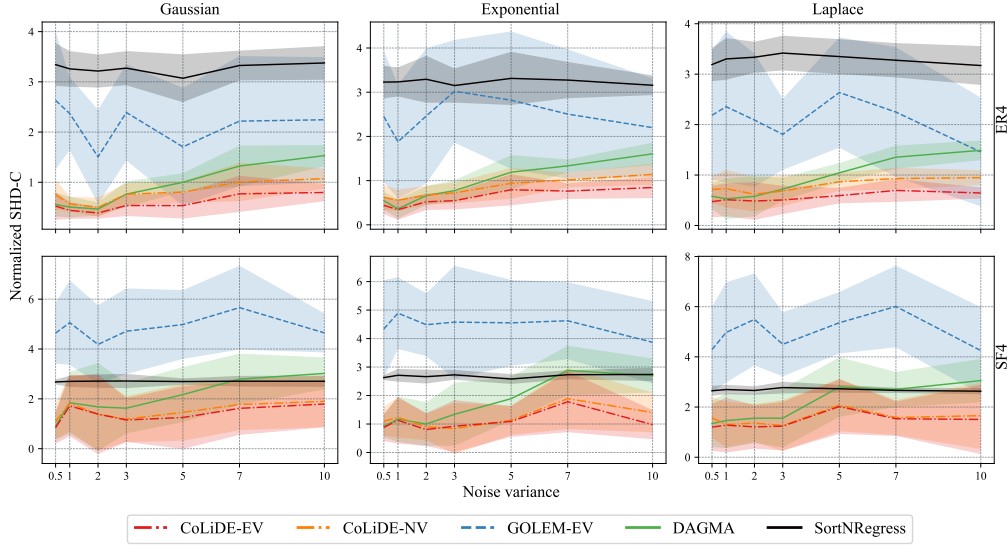

Figure 6: DAG recovery performance assessed for ER4 and SF4 graphs with 200 nodes, assuming equal noise variances. Each row represents a distinct noise distribution, and the shaded area depicts the standard deviation.

graphs, specifically those with 100 and 50 nodes, each with a nodal degree of 4. Utilizing the Linear SEM model, as detailed in Section 5, and assuming equal noise variances across nodes, we generated the respective data. The resulting normalized SHD and FDR are depicted in Figure 7, where the first three rows present results for 100-node graphs, and the subsequent rows showcase findings for 50-node graphs. It is notable that as the number of nodes decreases, GOLEM's performance improves, benefitting from a greater number of data samples per variable. However, even in scenarios with fewer nodes, CoLiDE-EV consistently outperforms other state-of-the-art methods, including GOLEM.

### E.3 ADDITIONAL METRICS FOR THE HETEROSCEDASTIC EXPERIMENTS

For the heteroscedastic experiments in Section 5, we exclusively displayed normalized SHD for various node counts and noise distributions. To ensure a comprehensive view, we now include normalized SID and FDR in Figure 8 and normalized SHD-C in Figure 9, representing the same graphs analyzed in Section 5. Figures 8 and 9 demonstrate the consistent superiority of CoLiDE-NV over other state-of-the-art methods across multiple metrics.

### E.4 DATA STANDARDIZATION IN HETEROSCEDASTIC SETTINGS

In simulated SEMs with additive noise, a phenomenon highlighted in (Reisach et al., 2021) indicates that the data generation model might inadvertently compromise the underlying structure, thereby unexpectedly influencing structure learning algorithms. Reisach et al. (2021) argue that, for generically sampled additive noise models, the marginal variance tends to increase along the causal order. In response to this observation, they propose a straightforward DAG learning algorithm named SortNRegress (see Appendix D.5), which we have integrated into our experiments to demonstrate the challenging nature of the task at hand. To mitigate the information leakage caused by such inadvertent model effects, Reisach et al. (2021) suggest standardizing the data of each node to have zero mean and unit variance.

We note that such standardization constitutes a non-linear transformation that can markedly impact performance, particularly for those methods developed for linear SEMs. For our experiments here, we utilized the same graphs presented in the heteroscedastic settings section of the paper but standardized the data. The recovery performance of the algorithms tests is evaluated using the normalized SHD and SHD-C metrics, as illustrated in Figure 10. Despite the overall degradation in per-

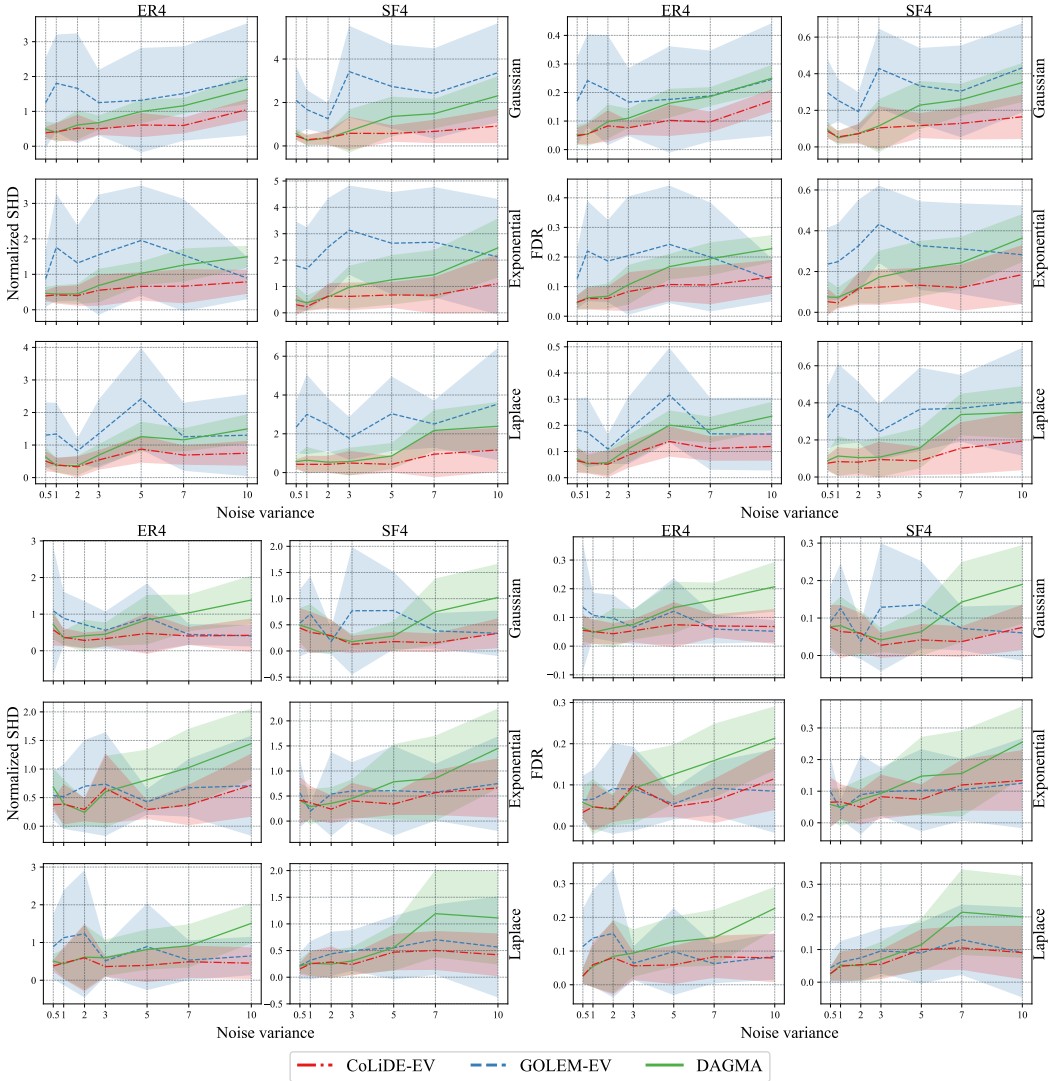

Figure 7: DAG recovery performance for graphs with 100 nodes (the first three rows) and 50 nodes. Each graph is designed with a nodal degree of 4, and equal noise variances are assumed across nodes. The initial two columns display normalized SHD, while the subsequent columns present FDR. The shaded areas indicate standard deviations.

formance resulting from such standardization, CoLiDE-NV continues to outperform state-of-the-art algorithms in this challenging task.

We also conducted a similar standardization on 10 small and sparse ER1 graphs, each comprising $d = 20$ nodes, with a substantial dataset of $n = 100,000$ i.i.d. observations. This time, we considered the greedy algorithm GES (see Appendix D.5) for comparisons. Prior to standardization, the performance of the algorithms in terms of SHD was as follows: CoLiDE-EV: 12.8; CoLiDE-NV: 14.9; and GES: 14.9. Post-standardization, the SHD values changed to CoLiDE-EV: 18.8; CoLiDE-NV: 19.5; and GES: 14.9. Notably, the performance of GES remained unaffected by standardization. While CoLiDE exhibited superior performance compared to others based on continuous relaxation (see Figure 10), it displayed some sensitivity to standardization, indicating that it is not as robust in this aspect as greedy algorithms. We duly acknowledge this as a limitation of our approach.

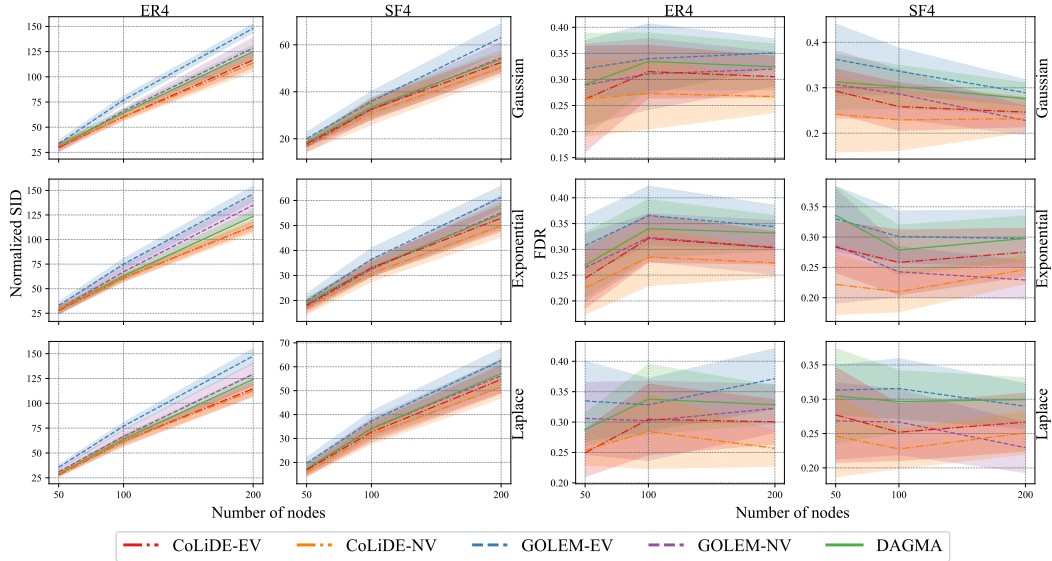

Figure 8: DAG recovery performance evaluated for ER4 and SF4 graphs with varying node counts under different noise distributions (each row). The figures display normalized SID (first two columns) and FDR (last two columns). We consider varying noise variances across nodes, and the shaded areas represent standard deviations.

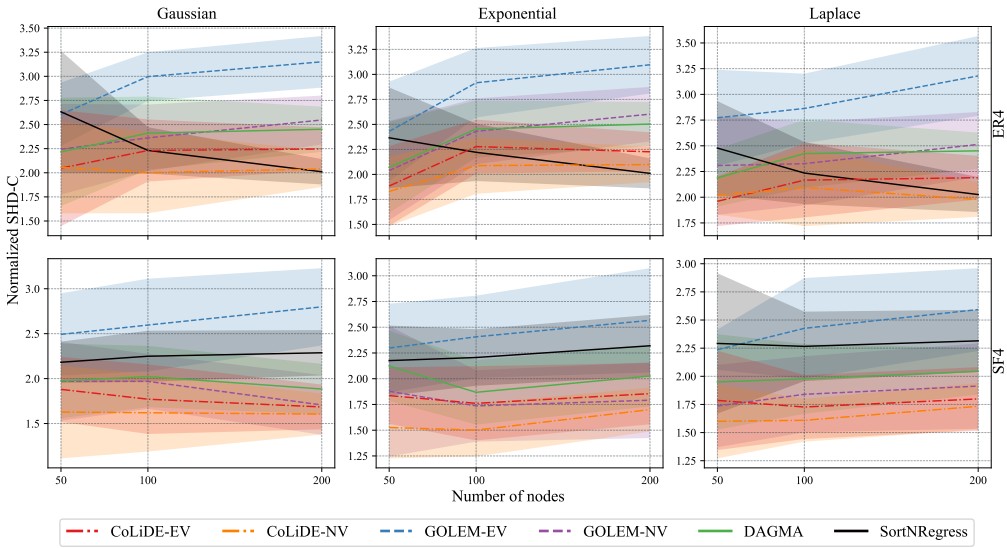

Figure 9: DAG recovery performance evaluated for ER4 and SF4 graphs with varying node counts under different noise distributions (each row). The figures display normalized SHD-C. We consider varying noise variances across nodes, and the shaded areas represent standard deviations.

## E.5 HIGH SNR SETTING WITH HETEROSCEDASTICITY ASSUMPTION

For the heteroscedastic experiments in Section 5, we discussed how altering edge weights can impact the SNR, consequently influencing the problem's complexity. In this section, we explore a less challenging scenario by setting edge weights to be drawn from $[-2, -0.5] \cup [0.5, 5]$, creating an easier counterpart to the heteroscedastic experiments in Section 5. We maintain the assumption of varying noise variances across nodes. The normalized SHD and FDR for these experiments are presented in Figure 11. As illustrated in Figure 11, CoLiDE variants consistently outperform other state-of-the-art methods in this experiment. Interestingly, CoLiDE-EV performs on par with

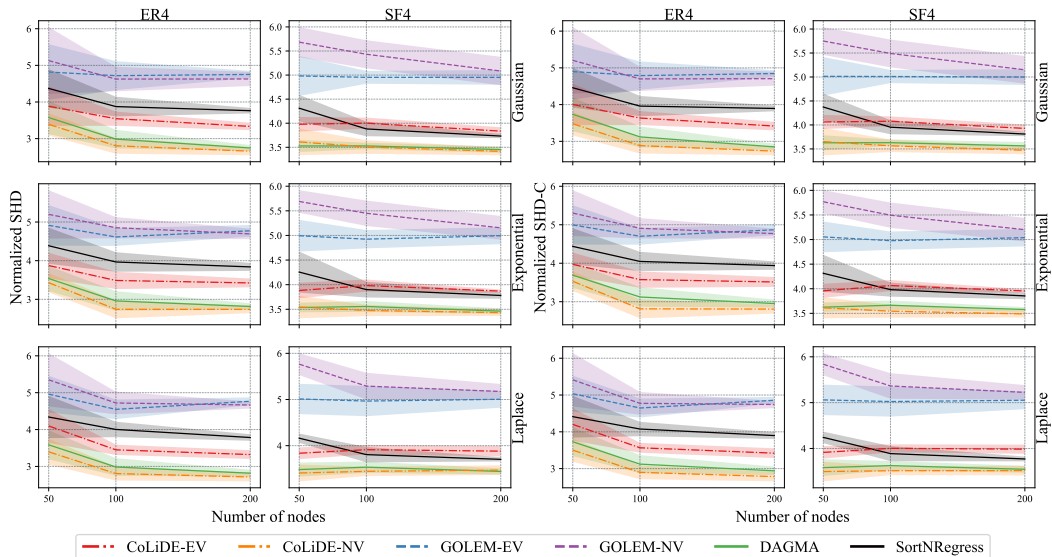

Figure 10: DAG recovery performance for ER4 and SF4 graphs when data standardization is applied, considering varying noise variances for each node. These experiments were conducted across different numbers of nodes and noise distributions (each row). The first two columns display normalized SHD, while the rest present normalized SHD-C. Shaded areas represent standard deviations.

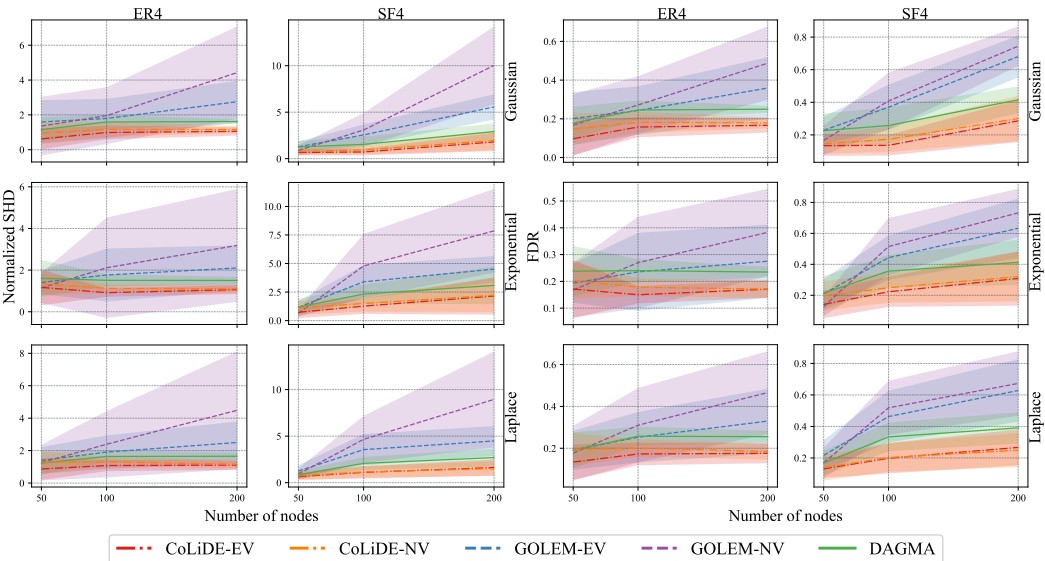

Figure 11: DAG recovery performance in a high SNR setting for ER4 and SF4 graphs, considering varying noise variances for each node. These experiments were conducted across different numbers of nodes and noise distributions (each row). The first two columns display normalized SHD, while the rest present FDR. Shaded areas represent standard deviations.

CoLiDE-NV in certain cases and even outperforms CoLiDE-NV in others, despite the mismatch in noise variance assumption. We attribute this to the problem's complexity not warranting the need for a more intricate framework like CoLiDE-NV in some instances.

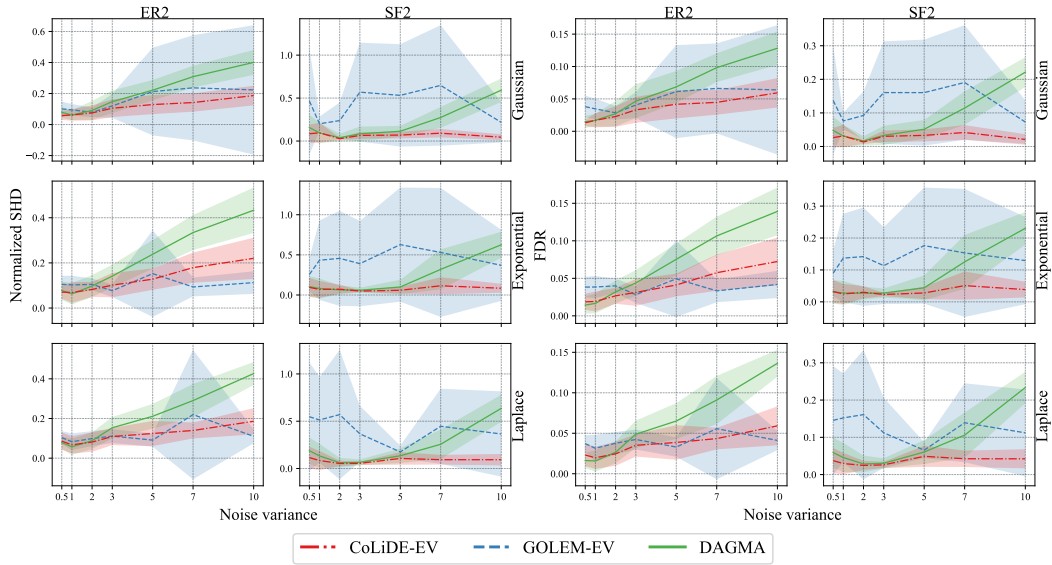

Figure 12: DAG recovery performance assessed on sparse graphs with a nodal degree of 2 and 500 nodes, assuming homoscedastic noise. The first two columns depict normalized SHD, and the following columns depict FDR. Shaded regions represent standard deviations.

## E.6 LARGER AND SPARSER GRAPHS

It is a standard practice to evaluate the recovery performance of proposed DAG learning algorithms on large sparse graphs, as demonstrated in recent studies (Ng et al., 2020; Bello et al., 2022). Therefore, we conducted a similar analysis, considering both Homoscedastic and Heteroscedastic settings, with an underlying graph nodal degree of 2.

**Homoscedastic setting.** We generated 10 distinct ER and SF DAGs, each consisting of 500 nodes and 1000 edges. The data generation process aligns with the setting described in Section 5, assuming equal noise variance for each node. We replicated this analysis across various noise levels and distributions. The performance results, measured in terms of Normalized SHD and FDR, are presented in Figure 12. CoLiDE-EV consistently demonstrates superior performance in most scenarios. Notably, GOLEM-EV performs better on sparser graphs and, in ER2 with an Exponential noise distribution and high variance, it marginally outperforms CoLiDE-EV. However, CoLiDE-EV maintains a consistent level of performance across different noise distributions and variances in homoscedastic settings.

**Heteroscedastic setting.** Similarly, we generated sparse DAGs ranging from 200 nodes to 500 nodes, assuming a nodal degree of 2. However, this time, we generated data under the heteroscedasticity assumption and followed the settings described in Section 5. The DAG recovery performance is summarized in Figure 13, where we report Normalized SHD and FDR across different noise distributions, graph models, and numbers of nodes. In most cases, CoLiDE-NV outperforms other state-of-the-art algorithms. In a few cases where CoLiDE-NV is the second-best, CoLiDE-EV emerges as the best-performing method, showcasing the superiority of CoLiDE's variants in different noise scenarios.

## E.7 DENSE GRAPHS

To conclude our analysis, we extended our consideration to denser DAGs with a nodal degree of 6.

**Homoscedastic setting.** We generated 10 distinct ER and SF DAGs, each comprising 100 and 200 nodes, assuming a nodal degree of 6. The data generation process was in line with the approach detailed in Section 5, assuming equal noise variance for each node. This analysis was replicated across varying noise levels and distributions. The performance results, assessed in terms of normalized SHD and FDR, are presented in Figure 14. CoLiDE-EV consistently demonstrates superior

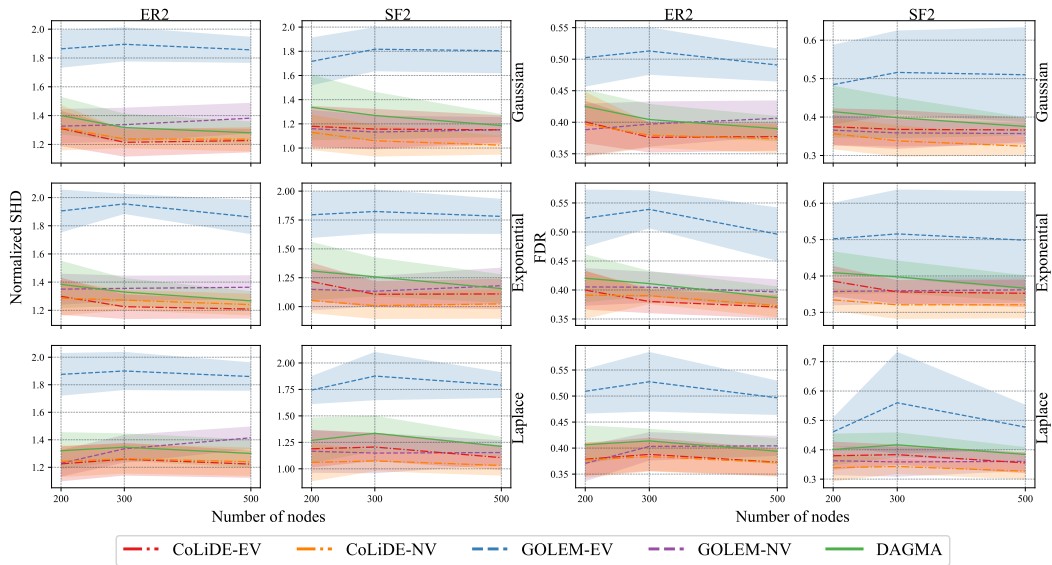

Figure 13: DAG recovery performance in large, sparse graphs with a nodal degree of 2, considering varying noise variances for each node. The experiments were conducted across different numbers of nodes and noise distributions (each row). The first two columns display normalized SHD, while the rest present FDR. Shaded areas represent standard deviations.

performance across all scenarios for DAGs with 100 nodes (Figure 14 - bottom three rows), and in most cases for DAGs with 200 nodes (Figure 14 - top three rows). Interestingly, in the case of 200-node DAGs, we observed that GOLEM-EV outperforms CoLiDE-EV in terms of normalized SHD in certain scenarios. However, a closer inspection through FDR revealed that GOLEM-EV fails to detect many of the true edges, resulting in a higher number of false positives. Consequently, the SHD metric tends to converge to the performance of CoLiDE-EV due to the fewer detected edges overall. This suggests that CoLiDE-EV excels in edge detection. It appears that CoLiDE-EV encounters challenges in handling high-noise scenarios in denser graphs compared to sparser DAGs like ER2 or ER4. Nonetheless, it still outperforms other state-of-the-art methods

**Heteroscedastic setting.** In this analysis, we also incorporate the heteroscedasticity assumption while generating data as outlined in Section 5. Dense DAGs, ranging from 50 nodes to 200 nodes and assuming a nodal degree of 6, were created. The DAG recovery performance is summarized in Figure 15, presenting normalized SHD (first two columns) and FDR across varying noise distributions, graph models, and node numbers. CoLiDE-NV consistently exhibits superior performance compared to other state-of-the-art algorithms in the majority of cases. In a few instances where CoLiDE-NV is the second-best, GOLEM-EV emerges as the best-performing method by a small margin. These results reinforce the robustness of CoLiDE-NV across a spectrum of scenarios, spanning sparse to dense graphs.

### E.8 INTEGRATING COLIDE WITH OTHER OPTIMIZATION METHODS

CoLiDE introduces novel convex score functions for the homoscedastic and heteroscedastic settings in (2) and (5), respectively. Although we propose an inexaxt BCD optimization algorithm based on a continuous relaxation of the NP-hard problem, our *convex* score functions can be readily optimized using other techniques. For instance, TOPO (Deng et al., 2023a) advocates a bi-level optimization algorithm. In the outer level, the algorithm performs a discrete optimization over topological orders by iteratively swapping pairs of nodes within the topological order of a DAG. At the inner level, given a topological order, the algorithm optimizes a score function. When paired with a convex score function, TOPO is guaranteed to find a local minimum and yields solutions with lower scores. The integration of TOPO and CoLiDE combines the former's appealing convergence properties with the latter's attractive features discussed at length in the paper.

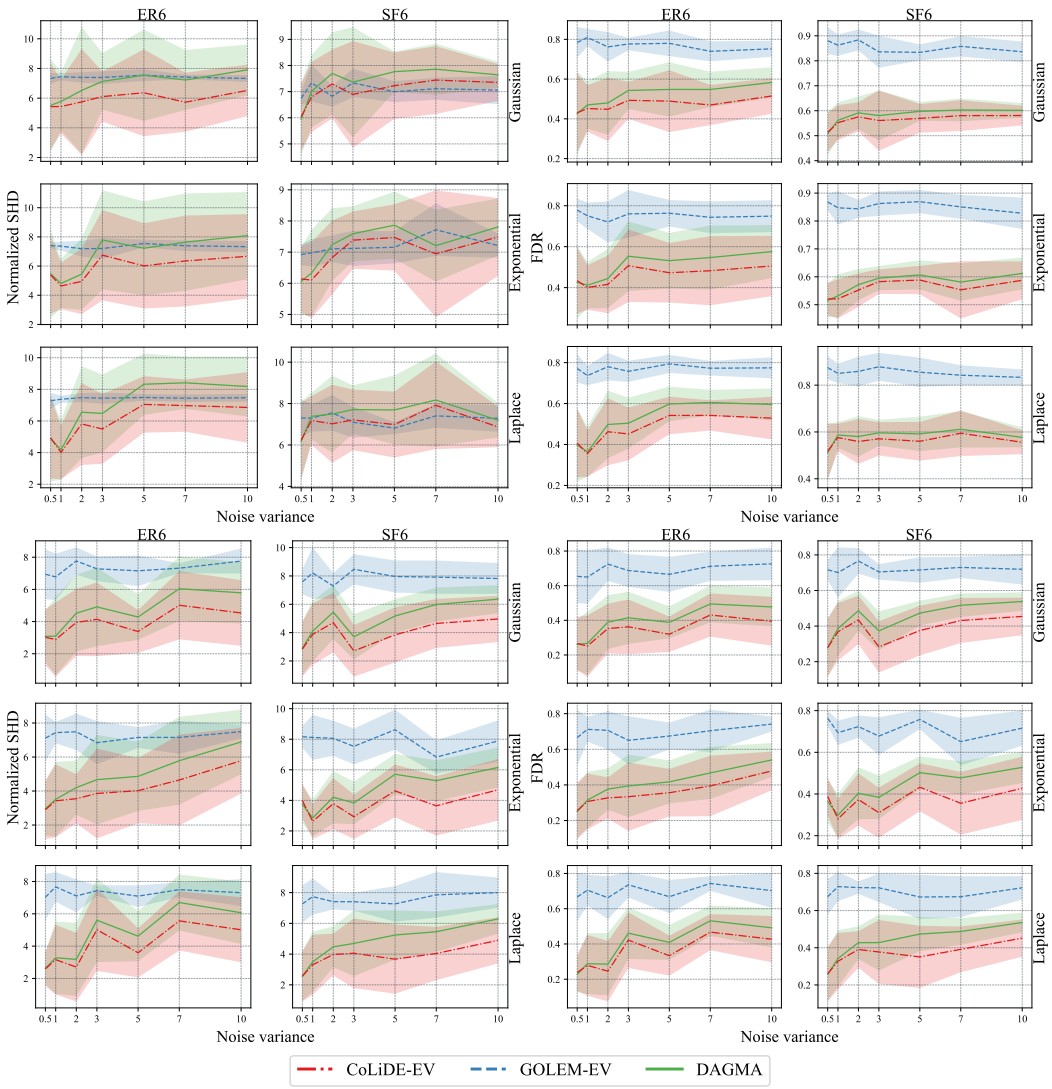

Figure 14: DAG recovery performance for graphs with 200 nodes (first three rows) and 100 nodes (bottom three rows). Each graph is constructed with a nodal degree of 6, assuming equal noise variances across nodes. The first two columns display normalized SHD, while the subsequent columns present FDR. Shaded areas represent standard deviations.

Using the available code for TOPO in https://github.com/Duntrain/TOPO, here we provide an initial assessment of this novel combination, denoted as CoLiDE-TOPO. We generate 10 different ER graphs with 50 nodes from a linear Gaussian SEM, where the noise variance of each node was randomly selected from the range $[0.5, 10]$. For the parameters of TOPO, we adhered to the values ($s_0 = 10$, $s_{small} = 100$, and $s_{large} = 1000$) recommended by Deng et al. (2023a). The results in Table 4 show that utilizing TOPO as an optimizer enables CoLiDE-NV to attain better performance, with up to 30% improvements in terms of SHD and for this particular setting.

Granted, the selection of the solver depends on the problem at hand. While discrete optimizers excel over smaller graphs, continuous optimization tends to be preferred for larger DAGs (with hundreds of nodes). Our experiments show that CoLiDE offers advantages in both of these settings.

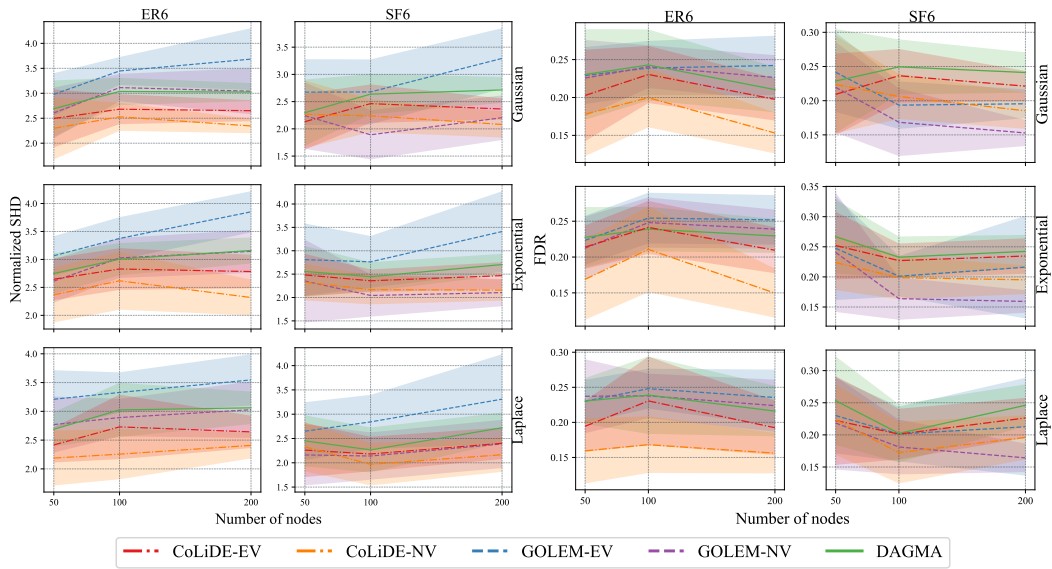

Figure 15: DAG recovery performance in dense graphs featuring a nodal degree of 6, considering differing noise variances for each node. The experiments were conducted across varying numbers of nodes and noise distributions (each row). The initial two columns illustrate normalized SHD, while the subsequent columns present FDR. Shaded areas depict standard deviations.

Table 4: DAG recovery results for 50-node ER4 graphs under heteroscedastic Gaussian noise.

|  | CoLiDE | | TOPO | CoLiDE-TOPO | |
| --- | --- | --- | --- | --- | --- |
|  | EV | NV |  | EV | NV |
| SHD | $100.7 \pm 29.6$ | $100.9 \pm 23.5$ | $128.4 \pm 38.0$ | $88.8 \pm 21.4$ | $\mathbf{66.9 \pm 15.4}$ |
| SHD-C | $102.4 \pm 30.0$ | $102.8 \pm 23.8$ | $131.1 \pm 37.3$ | $91.3 \pm 20.7$ | $\mathbf{68.8 \pm 15.4}$ |
| FDR | $0.26 \pm 0.10$ | $0.26 \pm 0.07$ | $0.36 \pm 0.08$ | $0.23 \pm 0.06$ | $\mathbf{0.16 \pm 0.04}$ |
| TPR | $0.69 \pm 0.06$ | $0.68 \pm 0.06$ | $0.72 \pm 0.06$ | $0.74 \pm 0.04$ | $\mathbf{0.74 \pm 0.04}$ |

## E.9 EXAMINATION OF THE ROLE OF SCORE FUNCTIONS

Here we conduct an empirical investigation on the role of score functions in recovering the underlying DAG, by decoupling their effect from the need of a (soft or hard) DAG constraint. Given a DAG $\mathcal{G}(\mathbf{W})$, the corresponding weighted adjacency matrix $\mathbf{W}$ can be expressed as $\mathbf{W} = \mathbf{\Pi}^{\top} \mathbf{U} \mathbf{\Pi}$, where $\mathbf{\Pi} \in \{0, 1\}^{d \times d}$ is a permutation matrix (effectively encoding the causal ordering of variables), and $\mathbf{U} \in \mathbb{R}^{d \times d}$ is an upper-triangular weight matrix. *Assuming we know the ground truth $\mathbf{\Pi}$* and parameterizing the DAG as $\mathbf{W} = \mathbf{\Pi}^{\top} \mathbf{U} \mathbf{\Pi}$ so that an acyclicity constraint is no longer needed, the DAG learning problem boils down to minimizing the score function $\mathcal{S}(\mathbf{W})$ solely with respect to $\mathbf{U}$

$$\min_{\mathbf{U}} \mathcal{S}\left(\mathbf{\Pi}^{\top} \mathbf{U} \mathbf{\Pi}\right). \tag{41}$$

Notice that (41) is a convex optimization problem if the score function is convex. Subsequently, we reconstruct the DAG structure using the true permutation matrix and the estimated upper-triangular weight matrix $\hat{\mathbf{W}} = \mathbf{\Pi}^{\top} \hat{\mathbf{U}} \mathbf{\Pi}$.

In this setting, we evaluate the CoLiDE-EV and CoLiDE-NV score functions in (2) and (5). In comparison, we assess the profile log-likelihood score function used by GOLEM (Ng et al., 2020). As a reminder, the GOLEM-NV score function tailored for heteroscedastic settings, is given by

$$-\frac{1}{2} \sum_{i=1}^{d} \log\left(\left\|\mathbf{x}_i - \mathbf{w}_i^{\top} \mathbf{X}\right\|_2^2\right) + \log\left(|\det(\mathbf{I} - \mathbf{W})|\right) + \lambda \|\mathbf{W}\|_1, \tag{42}$$

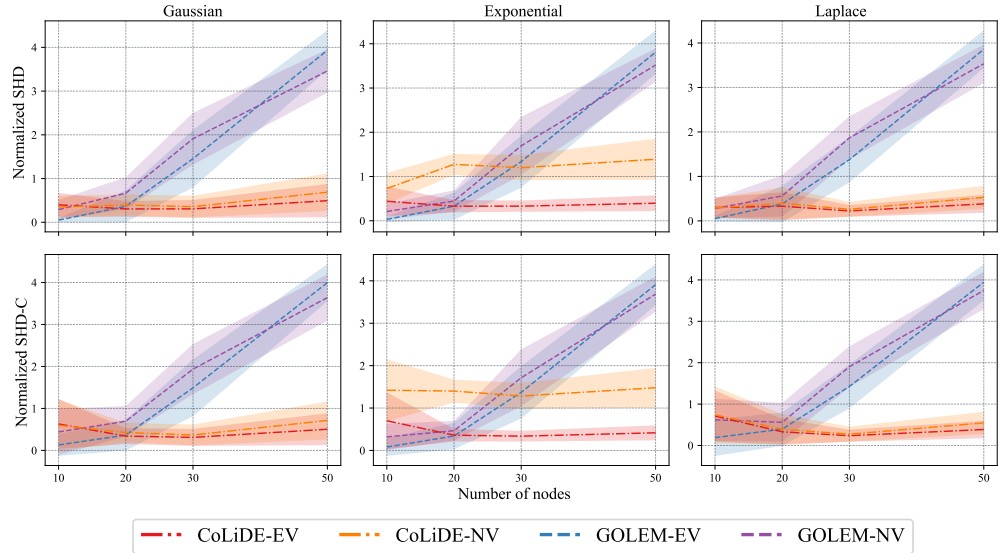

Figure 16: DAG recovery performance in ER4 DAGs under the assumption of known permutation matrices. The experiments were conducted for varying numbers of nodes and noise distributions (each column). The first row illustrates normalized SHD, while the subsequent row presents SHD-C. Shaded areas depict standard deviations.

where $\mathbf{x}_i \in \mathbb{R}^n$ represents node $i$'s data, and $\mathbf{W} = [\mathbf{w}_1, \dots, \mathbf{w}_d] \in \mathbb{R}^{d \times d}$. The GOLEM-EV score function, a simplified version of (42), is given by

$$-\frac{d}{2} \log \left( \left\| \mathbf{X} - \mathbf{W}^\top \mathbf{X} \right\|_F^2 \right) + \log \left( |\det(\mathbf{I} - \mathbf{W})| \right) + \lambda \|\mathbf{W}\|_1. \tag{43}$$

In all cases we select $\lambda = 0.002$ by following the guidelines in Ng et al. (2020).

It is important to note that the Gaussian log-likelihood score function used by GOLEM incorporates a noise-dependent term $-\frac{1}{2} \sum_{i=1}^d \log \sigma_i^2$, as discussed in Ng et al. (2020, Appendix C), which is profiled out during optimization. This noise-dependent term bears some similarities with our CoLiDE formulation. Indeed, after dropping the $\log (|\det(\mathbf{I} - \mathbf{W})|)$ term in (42) that vanishes when $\mathbf{W}$ corresponds to a DAG, the respective score functions are given by:

CoLiDE-NV: $\mathcal{S}(\mathbf{W}, \boldsymbol{\Sigma}) = \frac{1}{2n} \operatorname{Tr} \left( (\mathbf{X} - \mathbf{W}^\top \mathbf{X})^\top \boldsymbol{\Sigma}^{-1} (\mathbf{X} - \mathbf{W}^\top \mathbf{X}) \right) + \frac{1}{2} \sum_{i=1}^d \sigma_i + \lambda \|\mathbf{W}\|_1$

GOLEM-NV: $\mathcal{S}(\mathbf{W}, \boldsymbol{\Sigma}) = \frac{1}{2n} \operatorname{Tr} \left( (\mathbf{X} - \mathbf{W}^\top \mathbf{X})^\top \boldsymbol{\Sigma}^{-2} (\mathbf{X} - \mathbf{W}^\top \mathbf{X}) \right) + \frac{1}{2} \sum_{i=1}^d \log \sigma_i^2 + \lambda \|\mathbf{W}\|_1.$

While the similarities can be striking, notice that in CoLiDE the squared linear SEM residuals are scaled by the noise standard deviations $\sigma_i$. On the other hand, the negative Gaussian log-likelihood uses the variances $\sigma_i^2$ for scaling as in standard weighted LS squares. This observation notwithstanding, the key distinction lies in the fact that the negative Gaussian log-likelihood lacks convexity in the $\sigma_i$, making it non-jointly convex. In contrast, CoLiDE is jointly convex in $\mathbf{W}$ and $\boldsymbol{\Sigma}$ due to Huber's concomitant estimation techniques (see Appendix A.1), effectively convexifying the Gaussian log-likelihood; refer to Owen (2007, Section 5) for a more detailed discussion in the context of linear regression.

Going back to the experiment, we generate 10 different ER4 DAGs along with their permutation matrices. These graphs have number of nodes ranging from 10 to 50. Data is generated using a linear SEM under the heteroscedasticity assumption, akin to the experiments conducted in the main body of the paper. The normalized SHD and SHD-C of the reconstructed DAGs are presented in Figure 16. Empirical results suggest that the profile log-likelihood score function outperforms CoLiDE in smaller graphs with 10 or 20 nodes. However, as the number of nodes increases, CoLiDE (especially the CoLiDE-EV score function), outperforms the likelihood-based score functions.

