# OpenReview forum: "CoLiDE: Concomitant Linear DAG Estimation"
_ICLR.cc/2024/Conference — ICLR 2024 poster_

### Official Review · Reviewer_VEfR · 2023-10-31

**Soundness:** 2 fair
**Presentation:** 3 good
**Contribution:** 2 fair
**Rating:** 3
**Confidence:** 3

**Summary:**

The work proposes a new differentiable structure learning method for learning linear acyclic model that eliminates the assumption of equal error variances needed by several existing differentiable methods based on least squares. Building upon existing idea on smoothed concomitant lasso, the proposed method develops a regression-based score function that includes concomitant estimation of scale and decouples the sparsity parameter from the exogenous noise levels. Experiments with simulated and real-world datasets are provided.

**Strengths:**

The problem considered is highly relevant because it is important to relax the assumption of equal error variances to handle heteroscedastic noises.

**Weaknesses:**

The formulation (5) in the heteroscedastic setting lacks identification guarantee. It is unclear which specific settings it is theoretically correct for. For the linear Gaussian setting, one should use Gaussian likelihood, e.g., in GOLEM, while for linear non-Gaussian setting, one should use non-Gaussian likelihood, e.g., in NOTEARS-ICA.

There are some possible issues with the experiments, elaborated in the next section.

**Questions:**

- Does the method work after data standardization (see the study by Reisach et al. (2021)? Since the method is specifically for heteroscedastic setting, this experiment should be included to support the claim.
- For heteroscedastic Gaussian noise, the paper should compare the recovery results of Markov equivalence classes instead of DAGs, since the true DAG cannot be identified in theory.
- Regarding performance of DAGMA and GOLEM:
    - For DAGMA, did the authors try using the log-likelihood in the heteroscedastic setting? The authors of DAGMA paper consider such log-likelihood for nonlinear setting, but could be straightforwardly done for linear setting.
    -  For GOLEM, did the authors use the EV version to initialize the NV version, as suggested by their paper? Also, Section 5.1 says that GOLEM is based on profile-log-likelihood--I think a more straightforward comparison with Eq. (5) is their version without profiling.
    - Did the paper try to tune the hyperparameters for these two methods, since the settings considered here are quite different from their papers?
- What does "decouples the sparsity parameter from the exogenous noise levels" mean? I did not manage to find any elaboration or explanation of it.

---

> ### Author Response · Authors · 2023-11-18
> **Authors’ response to Reviewer VEfR (Part 1)**
>
> Thanks for your time and effort in reviewing our manuscript, as well as for finding the problem highly relevant due to the importance of considering heteroscedastic settings in linear DAG learning. Moreover, we appreciate your valuable suggestions to improve our numerical experiments in several ways, and your request for further elaboration on model identifiability issues. Point-by-point responses to your constructive comments and associated requests for changes follow, which we believe have led to an improved revised paper. We strive to improve our paper and will be happy to continue the scholarly discussion if any lingering issues remain.
>
> **Identifiability.** In the second paragraph of Section 1 we discuss different choices of score functions (likelihood and regression-based) and their relative merits. But with regards to guarantees, you bring up an excellent point that certainly benefits from additional elaboration. In (Loh and Buhlmann, 2014), the overarching assumption is that *the exogenous noise covariance matrix is known* up to a common scale across variables (e.g., in the homoscedastic setting), while CoLiDE also endeavors to estimate the exogenous noise variances. In the homoscedastic setting, the weighted least squares (LS) criterion therein and the CoLiDE-EV score function are equivalent, since the latter is obtained from weighted LS via a constant scaling and shift that will not affect the optimal $\mathbf{W}$ solution. Hence, the identifiability results in (Loh and Buhlmann, 2014) will carry over for both linear Gaussian and non-Gaussian SEMs, again, provided  the diagonal noise covariance matrix is known up to a constant factor.
>
> Now, the story is quite different in the more challenging heteroscedastic setting. As we now spell out in the introduction of the revised paper, for general linear Gaussian models the DAG structure is non-identifiable from observational data alone. Interestingly, just like GOLEM (Ng et al, 2020) and for general (non-identifiable) linear Gaussian SEMs, we can show that as $n\to\infty$ CoLiDE-NV probably yields a DAG that is *quasi-equivalent* to the ground truth graph. These guarantees for (heteroscedastic) linear Gaussian SEMs build on the interesting theoretical framework put forth in (Ghassami et al, 2020); see also Appendix C.
>
> Comments along these lines have been included in the revised manuscript; please check the paragraph immediately preceding Section 4.1.
>
> **Data standardization in heteroscedastic settings.** Following your suggestion, we have conducted an additional experiment involving standardized data in a heteroscedastic setting; please check Appendix E.4 in the revised manuscript. Despite an overall degradation in performance of all tested baselines resulting from such standardization (see Figure 9), CoLiDE-NV maintains its status as the superior method when compared to other state-of-the-art approaches.
>
> **Improved metrics over Markov equivalence classes.** This point is well taken. Following your suggestion, we have introduced a new metric termed SHD-C, defined in Appendix D.3 and computed as follows. We initially map both the estimated graph and the ground truth to their respective Completed Partially Directed Acyclic Graphs and subsequently calculate the SHD between them. Notably, SHD-C has been employed in (Ng et al, 2020); which also dealt with (non-identifiable) linear Gaussian SEMs. *SHD-C results are now reported in all relevant tables.* Furthermore, to complement the experiments presented in the main body of the paper, we have included Figures 5 and 8 in Appendix E.
>
> **Gaussian log-likelihood in the heteroscedastic setting.** Thanks again for this constructive feedback.  We conducted new experiments specifically aimed at assessing the performance of the CoLiDE score functions in comparison with the Gaussian log-likelihood (as utilized in GOLEM), *by decoupling their effect from the need of a (soft or hard) DAG constraint*. Please check Appendix E.9 in the revised manuscript for a detailed description of the experimental setting. As an ablation study, we believe this is even more informative than augmenting both score functions with, say, the DAGMA acyclicity function.  Our results indicate that for smaller DAGs (say with 10 nodes), the log-likelihood score functions outperform concomitant estimators. However, as the number of nodes increases, the score functions utilized in CoLiDE-EV and CoLiDE-NV exhibit superior performance (measured in terms of normalized SHD and SHD-C) when compared to the log-likelihood score functions of GOLEM-EV and GOLEM-NV.

---

> ### Author Response · Authors · 2023-11-18
> **Authors’ response to Reviewer VEfR (Part 2)**
>
> **Comparison with GOLEM.** Certainly, we always followed the guideline of initializing GOLEM-NV with the output of GOLEM-EV, as suggested in (Ng et al, 2020). To avoid any ambiguity, we have spelled this out in the revised manuscript; please check Appendix D.5. Our GOLEM experiments are based on the profiled likelihood approach in (Ng et al, 2020), and we used Ng et al’s own implementation. We carefully checked their paper and could not find a version of their method without profiling.
>
> **Hyperparameter tuning.** Our optimization methodology closely follows DAGMA. Given that the experimental settings are similar, for fair comparison we selected $\lambda=0.05$ and all other hyperparameters as in (Bello et al, 2022). To assess the robustness of said choice, we tested several other $\lambda$ values and found $0.05$ to be preferable. Importantly, CoLiDE decouples $\lambda$ from $\sigma$, eliminating the need for recalibration when there are variations in the noise levels. Clarifications along these lines have been included in the revised manuscript; please check Section 4.1. For other baselines, we chose the recommended hyperparameters as outlined in Appendix C.5.
>
> In closing, we would like to clarify that while some experimental settings considered here are different from those in the GOLEM and DAGMA papers, a few (see e.g., Table 1) are purposely identical to those therein; and CoLiDE’s advantage over its competitors is apparent.
>
> **Decoupling of sparsity regularization parameter and noise level.** We apologize for the lack of clarity in this key concept, which was probably due to our suboptimal exposition. We clarify this here. Optimal rates for lasso hinge on selecting $\lambda \asymp \sigma \sqrt{\log d / n}$; see e.g, (Li et al, 2020). However, the exogenous noise variance $\sigma^2$ is rarely known in practice. Interestingly, by virtue of the scaled residuals in concomitant lasso estimators then minimax optimality requires $\lambda\asymp \sqrt{\log d/n}$ (Li et al, 2020). This strategy renders $\lambda$ independent of $\sigma$, which is exactly what we mean when we say `decoupling the sparsity parameter from the exogenous noise levels’. Clarifying comments along these lines were included in the revised manuscript; see Sections 2, 4, and Appendix A.1.
>
> Thanks again for your valuable feedback that has contributed to an improved revised manuscript, and we look forward to addressing any further concerns that may arise.

---

> ### Author Response · Authors · 2023-11-22
> **Follow-up**
>
> We would be happy to answer any follow-up questions. Thanks!

---

> ### Comment · Reviewer_VEfR · 2023-11-23
>
> Thanks for the response and clarifications. Some of my concerns have been addressed. My comments are as follows.
>
> **Data standardization in heteroscedastic settings**: The experiments in Appendix E.4 show that there is a huge drop in the performance of the proposed method (despite outperforming existing differentiable methods). Since the goal of the method is to handle heteroscedastic scenarios, the method should ideally be similar to GES/PC that handle heteroscedastic case and is less sensitive to data standardization, which is not the case here. This indicates that there are still challenges for the proposed method to handle heteroscedastic scenarios.
> - For example, the authors could consider a simpler Gaussian setting with 20 nodes, ER-1 graphs, and a large number of samples (e.g. $100000$), and further compare all methods, in addition to GES/PC, after and before standardizing the data. If, in this case, the performance of the proposed method becomes much worse after standardizing data, while GES/PC have a relatively stable performance, then it may mean that the heteroscedastic scenarios have not been fully resolved.
>
> **Gaussian log-likelihood**: After looking at the derivation of GOLEM, I notice that Eq. (5) is similar to the Gaussian likelihood; see the first few lines of p. 15 in the GOLEM paper (before profiling the noise variance). The determinant vanishes when $B$ is acyclic. Therefore, after this determinant vanishes, the difference of (unprofiled) Gaussian likelihood with Eq. (5) seems to be that the latter involves $\sum_i \sigma_i$, while the former involves $\sum_i \log \sigma_i^2$ (similarly for the EV version). There is also minor difference in the first term of Eq. (5). Could the authors explain more about this similarity/connection (e.g. why likelihood involves a "log" function but Colide does not), and how Eq. (5) is related to Gaussian likelihood? (E.g., are they "equivalent" in some sense?) Why one is better than the other specifically in the Gaussian case?
>
> Given such similarity, it would be insightful to compare to the unprofiled Gaussian likelihood (see first few lines of p. 15 in the GOLEM paper), as this is a more direct comparison with Colide (so that both of them do not profile any parameter).
>
> **Identifiability**: I find the discussion in Appendix C to be less rigorous. For example, it states $\lambda=0$, $\mu_k\rightarrow\infty$, and Eq. (40); it then states that the rest of the proof is similar to (Ng et al., 2020). However, after trying to look at the result/proof in (Ng et al., 2020), sparsity should still be needed for the proof, which contradicts with $\lambda = 0$. Also, not sure if I miss something: isn't $\mu_k\rightarrow\infty$ contradictory with the choice of $\mu_k\rightarrow 0$ for barrier method used by Colide and DAGMA? I find it hard to understand why one brings $\mu_k$ to infinity.

---

### Official Review · Reviewer_wJd3 · 2023-10-31

**Soundness:** 4 excellent
**Presentation:** 3 good
**Contribution:** 3 good
**Rating:** 8
**Confidence:** 5

**Summary:**

The paper introduces a new continuous optimization problem for DAG learning. It leverages results from the concomitant scale estimation literature to learn a weighted adjacency matrix while estimating the scale of the exogenous noise variables. The experiments clearly show that optimizing the new objective (by inexact block coordinate descent) instead of the original l1-regularized objective of DAGMA results in better estimation of the graph across various settings.

**Strengths:**

1. The paper tackles an important problem of interest to the general ICLR community.

2. The proposed regularization is general enough that it can be plugged in many of the continuous optimization problems recently proposed for learning DAGs. The work's impact is hence potentially high as it could improve performance of many state-of-the-art methods.

3. The paper is generally well presented. Its claims are well supported by an extensive empirical analysis that illustrates the DAG recovery capabilities of the method on several settings and for noise estimation.

**Weaknesses:**

1. Although the adjacency matrix $W$ can be efficiently updated with stochastic gradient steps, the closed-form for the noise scale is evaluated on the full data because it is not decomposable. This makes the method scale poorly to big data. This limitation should be highlighted in the text or an efficient approximation could be discussed and empirically evaluated.

2. It is not clear how Problem 2 is obtained, i.e., under which assumptions the noise-dependent terms appear in the objective. It would be useful to report such derivation in the appendix. This would in particular allow for verifying if the sparsity inducing term $||W||_1$ can be replaced by the score-equivalent term $||W||_0$, used e.g. in [Brouillard et al. 2020, Zantedeschi et al. 2023], without loss in noise estimation performance.

3. The paper does not describe how $\lambda$ was tuned. In Section 4.1 it is only mentioned that it was "empirically determined".

4. Sortnregress should be reported also in Figure 1. Currently the text reports sortnregress results for two values of the noise scale and for a single graph type, but it wouldn't hurt the readability of Figure 1 to add all the results for that baseline for all the settings. Plotting such results would allow to clearly see which settings are trivial and which are of interest.

I would be inclined to increase my rating if these points are addressed.

**Questions:**

(minor) There is a sign typo in the second-last equation of page 14.

---

> ### Author Response · Authors · 2023-11-18
> **Authors’ response to Reviewer wJd3**
>
> Thanks for your time and effort in reviewing our manuscript, as well as for finding the problem of broad interest to the ICLR community, the empirical analysis extensive, and the paper well written. We are glad to hear you believe this work’s impact can be potentially high, with benefits that could permeate to other continuous optimization approaches to DAG learning. Even more, we believe our ideas could also contribute to order-based methods, and in the revised paper we conduct a preliminary test based on valuable feedback provided by Reviewer xkAR.   Point-by-point responses to your comments and associated requests for changes follow. We strive to improve our paper and will be happy to continue the scholarly discussion if any outstanding issues remain.
>
> **Decomposability of scale estimator.** This is an excellent point. It is true that (unlike other approaches) the adjacency matrix $\mathbf{W}$ can be efficiently updated with stochastic gradient steps using individual samples or mini-batches, but *the same holds true for the noise scale estimators* in (4) and (6). This follows because these estimators depend on $\operatorname{cov}(\mathbf{X}) := \frac{1}{n} \mathbf{X} \mathbf{X}^{\top}$, and the sample covariance matrix can be updated *recursively* from individual samples or mini-batches. In this sense, the noise scale estimators are decomposable as well and can be implemented from partial or streaming data using, e.g., online algorithms that we sketch in the revised manuscript. Please check the `Online updates’ sections under Appendices B.1 and B.2. A full exploration of this exciting direction is left as future work.  Moreover, in the revised paper we explicitly state that CoLiDE’s score function is fully decomposable; see the discussion following (3).
>
> **Noise-dependent term in the score function.** The origin of concomitant scale estimators can be traced back to (Huber, 1981), who proposed *convex* criteria to jointly estimate linear regression coefficients along with the noise standard deviation $\sigma$. A first key ingredient is to note that for convex loss $\rho(u)$ (e.g., $\rho(u)=u^2$ in ordinary least squares), then it follows that $\rho(u/\sigma)\times \sigma$ is jointly convex in $(u,\sigma)$. Second, Huber pointed out that the inclusion of the (convexity preserving) linear term $d\sigma/2$ as in (2), yields an estimator $\hat{\sigma}^2$ that is consistent under Gaussianity [it is actually the maximum likelihood estimator; see (18)]. A clarifying comment on the reason behind  the addition of this linear term in the objective function has been included in the revised paper; please check the discussion following (2).
>
> It should thus be clear that the sparsity regularization (which only acts on $\mathbf{W}$) should not affect the *form* of the noise scale estimator $\hat{\sigma}$. It may affect noise estimation *performance* though,  since $\hat{\sigma}$ depends on $\hat{\mathbf{W}}$ [cf. (4)].  One could certainly use the edge cardinality function $||\mathbf{W}|||_0$ as a penalty, but we opted against it so as to retain convexity of the score function $\mathcal{S}(\mathbf{W},\sigma)$.
>
> **Hyperparameter tuning.** Our optimization methodology closely follows DAGMA. Given that the experimental settings are similar, for fair comparison we selected $\lambda=0.05$ as in (Bello et al, 2022). To assess the robustness of said choice, we tested several other $\lambda$ values and found $0.05$ to be preferable. Importantly, CoLiDE decouples $\lambda$ from $\sigma$, eliminating the need for recalibration when there are variations in the noise levels. Clarifications along these lines have been included in the revised manuscript; please check Section 4.1.
>
> **SortNRegress in Figure 1.** Following your suggestion, we have incorporated results for SortNRegress into Figure 1. We agree that plotting these results helps better convey the non-trivial nature of the task at hand.
>
> We have also fixed the typo in the gradient expression (16). Good catch!
>
> Thanks again for your valuable feedback that has contributed to an improved revised manuscript, and we look forward to addressing any further concerns that may arise. We appreciate your willingness to increase your rating.

---

> > ### Comment · Reviewer_wJd3 · 2023-11-22
> >
> > Thank you for addressing the raised points and for providing clarification. I find the arguments generally convincing.
> >
> > The only minor issue still standing from my side is the lack of experimentation with the online updates of the scale estimator, as described in the revision of the paper. As these updates rely on sufficient statistics, their variance might affect performance and it would have been valuable to empirically study its effect, especially because one of the claims of the paper is to have a fully decomposable loss well-suited to stochastic optimization, unlike existing methods that also estimate the noise scale.

---

> > > ### Author Response · Authors · 2023-11-23
> > > **Thanks for your additional feedback**
> > >
> > > We are glad to hear you found the arguments in our responses convincing.
> > >
> > > And thanks for keeping us honest about our claims. To address this lingering minor issue, we have included and additional sentence in Section 6 - Concluding Summary, Limitations, and Future Work of the revised manuscript, which reads ``Although CoLiDE's decomposability is a demonstrable property, experimental results are needed to fully assert the practical value of stochastic optimization.'' We would appreciate if, after this clarification, you would be willing to increase your score as originally intended.

---

> ### Author Response · Authors · 2023-11-22
> **Follow-up**
>
> We would be happy to answer any follow-up questions. Thanks!

---

### Official Review · Reviewer_xkAR · 2023-11-02

**Soundness:** 2 fair
**Presentation:** 4 excellent
**Contribution:** 2 fair
**Rating:** 6
**Confidence:** 4

**Summary:**

The paper studies the problem of DAG structure learning from a score-based viewpoint for linear models.
The authors propose a new score function that also estimates the noise levels and experimentally show that it can lead to better accuracies by leveraging recent advances in continuous non-convex characterizations of DAGs.

**Strengths:**

* The paper is clearly written and the contributions are easy to digest.
* The proposed score leads to structure improvements w.r.t. sota methods.

**Weaknesses:**

* Significance: The paper considers only linear models, hindering the significance of the proposed loss function.
* Novelty: The authors borrow ideas from concomitant lasso, and straightforwardly apply it to the score function for DAG learning. While it is totally okay with borrowing ideas from prior work, it feels that this is indeed the only technical contribution of the paper. The optimization part feels identical to prior work expect for the extra noise terms.

**Questions:**

* With respect to my point in the weaknesses section, in my opinion, it would be more enlightening to show that the proposed score function leads to identify the true underlying DAG. The current contribution feels like just "another score function" with no guarantees of identifiability. The non-equal noise variances was also studied in Loh and Buhlmann (2014) where they proposed a weighted LS that would lead to identifiability of the true DAG, this weighted LS depends on the noise levels as well, I wonder if jointly optimizing such objective would also lead to accuracy improvements.

* I wonder if the authors experimented with non-linear models as well?  Given that I would consider this work to be "empirical", it would be good to use these ideas into nonlinear models as well.

* I will also note, a recent method called TOPO by Deng et al. (2023) "Optimizing NOTEARS objectives via topological swaps" shows improvements in structure estimation for score-based methods. Their theory suggests that given a convex score (as in this paper) their optimization algorithm would guarantee a local optimum. It would be interesting to see if using the proposed convex score + TOPO can  obtain even more accurate DAGs, specially for non-equal variances. Finally, the same authors have provided initial insights into global optimality of continuous DAG learning methods which can also help to motivate this line of work in the continuous-constrained framework, see Deng et al (2023) "Global Optimality in Bivariate Gradient-based DAG Learning".

---

> ### Author Response · Authors · 2023-11-18
> **Authors’ response to Reviewer xkAR (Part 1)**
>
> Thanks for your time and effort in reviewing our manuscript, as well as for recognizing the contributions and CoLiDE’s improved performance over state-of-the-art approaches. We are glad to hear you found the paper to be clearly written. Moreover, we appreciate your valuable suggestions to further improve DAG structure estimation, by bringing to bear exciting optimization advances along with the CoLiDE score function. Point-by-point responses to your comments and associated requests for changes follow. We strive to improve our paper and will be happy to continue the scholarly discussion if any lingering issues remain.
>
> **Significance.** This point is well taken. Indeed, as even the paper title indicates the scope of this study is admittedly limited to DAG structure learning from observational data adhering to *linear* SEMs. We are the first ones to acknowledge this as one of the limitations of this work (see Section 6). While it is clear that CoLiDE *can* be extended to non-linear SEMs, we feel it is not prudent to include additional experiments in what is already a fully-packed paper. This issue notwithstanding, we contend that the linear case is still a significant and challenging problem, particularly in the heteroscedastic setting dealt with where. CoLiDE, our novel score-based approach exhibits significant improvements relative to existing state-of-the-art methods, across a broad swath of metrics and settings spanning different graph ensembles, exogenous noise distributions (with and without equal variances across variables), and even a real dataset that has been widely used for benchmarking. In terms of potential broader impacts, our novel convex score function incorporates concomitant estimation of scale and has wide applicability when combined with (or used in place of) conventional loss functions in other state-of-the-art DAG learning methods. Finally, while this work is framed within the continuous constrained relaxation paradigm to linear DAG learning, preliminary experiments with TOPO (thanks for this valuable suggestion!) show that CoLiDE also benefits order-based methods.
>
> **Novelty.** This work contributes several innovations to the timely field of learning DAG structure from observational data. First, we propose a new convex score function for sparsity-aware learning of linear DAGs, which incorporates concomitant estimation of scale parameters to enhance DAG topology inference using continuous first-order optimization. To the best of our knowledge, this is the first time that ideas from concomitant scale estimation permeate benefits to DAG learning. Second, we develop scalable (inexact) block-coordinate descent iterations to jointly estimate the DAG topology along with the exogenous noise levels – both in homoscedastic and heteroscedastic settings and with a marginal increase in complexity over structure estimation alone. Unlike some other existing approaches, we also show our score function is fully decomposable across samples, thus facilitating mini-batch based optimization if needed for scalability. Third, in existing methods the sparsity regularization parameter depends on the unknown exogenous noise levels, making the calibration challenging. CoLiDE effectively removes this coupling, leading to minimum (or no) recalibration effort across diverse problem instances – hence a valuable form of robustness that leads to tangible computational savings. Finally, through a comprehensive experimental evaluation protocol we demonstrate CoLiDE outperforms state-of-the-art methods without incurring added complexity, especially when the DAGs have several hundreds of nodes and the noise level profile is heterogeneous.

---

> ### Author Response · Authors · 2023-11-18
> **Authors’ response to Reviewer xkAR (Part 2)**
>
> **Identifiability.** This is an excellent point that certainly benefits from additional elaboration. In (Loh and Buhlmann, 2014), the overarching assumption is that *the exogenous noise covariance matrix is known* up to a common scale across variables (e.g., in the homoscedastic setting), while CoLiDE also endeavors to estimate the exogenous noise variances. In the homoscedastic setting, the weighted least squares (LS) criterion therein and the CoLiDE-EV score function are equivalent, since the latter is obtained from weighted LS via a constant scaling and shift that will not affect the optimal $\mathbf{W}$ solution. Hence, the identifiability results in (Loh and Buhlmann, 2014) will carry over for both linear Gaussian and non-Gaussian SEMs, again, provided  the diagonal noise covariance matrix is known up to a constant factor. For the aforementioned reasons, we respectfully believe there will be no added benefit in combining CoLiDE-EV with the weighted LS loss in (Loh and Buhlmann, 2014).
>
> Now, the story is quite different in the more challenging heteroscedastic setting. As we now spell out in the introduction of the revised paper, for general linear Gaussian models the DAG structure is non-identifiable from observational data alone. Interestingly, just like GOLEM (Ng et al, 2020) and for general (non-identifiable) linear Gaussian SEMs, we can show that as $n\to\infty$ CoLiDE-NV probably yields a DAG that is *quasi-equivalent* to the ground truth graph. These guarantees for (heteroscedastic) linear Gaussian SEMs build on the interesting theoretical framework put forth in (Ghassami et al, 2020); see also Appendix C.
>
> Comments along these lines have been included in the revised manuscript; please check the paragraph immediately preceding Section 4.1.
>
> **Combining the CoLiDE score function with TOPO.** Thanks for bringing both relevant works by Deng et al to our attention, which we now cite and discuss in the revised manuscript. Following your excellent suggestion, we have implemented a new method that combines the convex CoLiDE score functions along with TOPO. Our encouraging preliminary results in Table 4 show that utilizing TOPO as an optimizer enables CoLiDE-NV to attain better performance, with up to 20% improvements in terms of SHD and for the particular setting tested (10 different ER graphs with 50 nodes from a linear Gaussian SEM, where the noise variance of each node was randomly selected from the range $[0.5, 10]$). Additional metrics are presented in Table 4 for a comprehensive evaluation. Please check Appendix E.8 for additional details of this new experiment. We have appended our CoLiDE-TOPO implementation to the anonymized code that we made publicly available as supplementary material accompanying this submission.
>
> In closing, we note that the selection of the solver depends on the problem at hand. While discrete optimizers excel over smaller graphs, continuous optimization tends to be preferred for larger DAGs (with hundreds of nodes). Our experiments show that CoLiDE offers advantages in both of these settings, which reinforces the significance of the proposed score function for linear DAG learning.
>
> Thanks again for your valuable feedback that has contributed to an improved revised manuscript, and we look forward to addressing any further concerns that may arise.

---

> ### Author Response · Authors · 2023-11-22
> **Follow-up**
>
> We would be happy to answer any follow-up questions. Thanks!

---

> > ### Comment · Reviewer_xkAR · 2023-11-23
> >
> > I thank the authors for their response, it has generally clarified my questions. The revision taking other reviewers' points has led to a stronger version. I am adjusting my score accordingly.

---

> > > ### Author Response · Authors · 2023-11-23
> > > **Thanks**
> > >
> > > We are glad to hear our responses clarified your questions and we agree the paper is stronger now. We appreciate your feedback and willingness to adjust your score.

---

### Meta-Review · Area_Chair_3z9Q · 2023-12-10

**Metareview:**

This paper presents a method for estimating DAGs from observational data under a linear structural equation model. It proposes a convex score function for sparsity-aware learning of linear DAGs which addresses the limitations of previous approaches: (1) expensive penalty terms and (2) homoscedasticity. Thus, by drawing ideas from the concomitant lasso, the paper deals with the heteroscedastic setting.

The reviewers found this paper addresses an important problem in ML and is clearly written. The proposed regularisation is general enough and the claims are supported by the empirical evaluation, outperforming the SOTA. The main issues raised by the reviewers concerned the significance of the approach (as it only deals with the linear case), the novelty of the approach and expanding on identifiably issues. As described in the authors' rebuttal, I believe the problem is challenging enough in the heteroscedastic setting and it is OK to leave the nonlinear case to further work. I also think that the paper presents a contribution significant enough regardless of how many new methodological tricks were developed, although the authors do mention that there were also contributions there. In terms of identifiability, the authors also provide some quasi-equivalent (to the ground truth) results.

There are other issues that remain unsolved, such as the claimed decomposability of the approach. Although, in principle, the approach is decomposable (and hence amenable to mini-batch learning), it seems that experiments were not carried out for this in terms of the updates of the noise scales. Also other concerns, mainly by Reviewer VEfR are not completely solved (e.g.), in terms of data standardisation.

With the above, I believe the paper presents a contribution worth presenting at ICLR and recommend acceptance. However, I encourage the authors to include results using the fully decomposable version of the method. Regardless of the results, these can still be analysed. The authors should also discuss the limitations of the approach more comprehensively (with regards to Reviewer VEfR concerns).

**Justification For Why Not Higher Score:**

There were still important issues raised by the re not fully solved in the rebuttal.

**Justification For Why Not Lower Score:**

good paper, deals with the harder heteroscedastic setting and good empirical results.

---

### Decision · Program_Chairs · 2024-01-16

Accept (poster)